# Adoption of Road Water Harvesting Practices and Their Impacts: Evidence from a Semi-Arid Region of Ethiopia

**Kebede Manjur Gebru** [1,*] , **Kifle Woldearegay** [2] , **Frank van Steenbergen** [3] , **Aregawi Beyene** [1] , **Letty Fajardo Vera** [3] , **Kidane Tesfay Gebreegziabher** [1] and **Taye Alemayhu** [3]

1   Department of Rural Development &Agricultural Extension, Mekelle University,
    P.O. Box 231 Mekelle, Ethiopia; aregawibeyene04@gmail.com (A.B.); kidanetesfay21@gmail.com (K.T.G.)
2   School of Earth Sciences, Mekelle University, P.O. Box 231 Mekelle, Ethiopia; kiflewold@yahoo.com
3   MetaMeta Research, Postelstraat 2, 5211 EA's-Hertogenbosch, The Netherlands;
    fvansteenbergen@metameta.nl (F.v.S.); lfajardovera@metameta.nl (L.F.V.); taye@metameta.nl (T.A.)
*   Correspondence: solomonmanjur@gmail.com or kebede.manjur@mu.edu.et; Tel.: +251-914731423

**Abstract:** In the drylands of Ethiopia, several road water harvesting practices (RWHP) have been used to supplement rain-fed agriculture. However, factors affecting adoption of RWHP and their impacts were not studied systematically. Understanding the factors influencing the adoption of RWHP for sustainable agricultural intensification and climate resilience is critical to promoting such technologies. This paper investigates the impacts of using rural roads to harvest rainwater runoff and the factors causing farmers to adopt the practice. Road water harvesting is considered a possible mechanism for transformative climate change adaptation. By systematically capturing rainfall with rural road infrastructure, rain-related road damage is reduced, erosion and landscape degradation due to road development is lessened, and farm incomes increase due to the beneficial use of harvested water, resulting in an increased climate change resilience. This paper uses a binary probit model and propensity score matching methods based on a household survey of 159 households and 603 plots. The results of the probit model show that the education level of the household, family labor, access to markets, and distance of the farming plot from the farmer's dwelling are statistically significant in explaining farmers' adoption of RWHP in the study area. The casual impact estimation from the propensity score matching suggests that RWHP has positive and significant impacts on input uses (farmyard manure and fertilizer), crop yield, and farm income among the sample households.

**Keywords:** adoption; farmyard manure; fertilizer; income; Northern Ethiopia; road water harvesting; yield

## 1. Introduction

Ethiopia's estimated population of over 100 million makes it the second most populous country in Africa. Ethiopia is agro-ecologically and ethnically diverse, and predominantly agrarian [1]. Rain-fed agriculture plays a pivotal role in the national economy, accounting for 36% of the Gross Domestic Product (GDP), 70% of export earnings, 76% of the livelihood of the country's workforce, and nearly 80% of employment in rural areas [2]. However, the agriculture sector in the dryland areas of the country suffers from drought-induced moisture stress events [3]. Dryland areas in Ethiopia cover three-quarters of the country's landmass, and one-third of the population [4]. Hence, changes in the biophysical environment, such as rainfall fluctuations in dryland areas, easily destabilize the national economy [5,6].

Dryland areas are characterized by high intra- and inter-seasonal rainfall variability [7]. Agriculture in these areas is considered a risky and challenging undertaking, as rainfall is relatively inadequate with

high spatial and temporal variability, manifested in prolonged dry spells during the growing season with the net effects of agricultural yield damage and food insecurity [2,3]. This often results in reduced life expectancy or migration for the rural poor who depend on rain-fed agriculture [7]. Furthermore, due to the high risks associated with water availability for plant growth, farmers in semi-arid areas tend to refrain from using productivity-enhancing inputs. This, together with the fluctuations in yields, makes it hard for subsistence farmers in semi-arid areas to stay and sustain their livelihoods in agriculture [8].

Dramatic consequences of climate changes have recently been experienced in eastern African countries (such as Ethiopia, Somalia, and Kenya), where many parts of the region have serious floods, followed by drought and then floods again [9]. Inadequate and extreme fluctuations in the amount of water available in the root-zone is a major constraint to the productivity and profitability of agriculture, resulting in a poverty trap for smallholder farmers. Evidence shows that one drought occurrence in a decade can lower a country's national GDP by 10% and increase poverty by 14% [6].

This negative spiral of insecurity and inability to buffer water resources is a significant problem. A virtuous cycle of green future farming is needed. Depending on the local context in dryland areas, this can be achieved through different techniques. Several studies have been conducted on the effect of rain water harvesting (RWH) in arid and semiarid regions around the world [6,7,10,11]. RWH is a well-known practice designed to improve water security and agricultural production. RWH includes different methods for inducing, collecting, storing, and conserving local surface runoff. Climate change and increases in water demand have renewed interest in the management of RWH in drylands. Increases in drought frequency and extreme precipitation events emphasize the role of RWH in improving water security. One part of RWH technique is harvesting runoff water from roads, which is termed as road water harvesting practices (RWHP). Road water harvesting practices include diverting runoff (from roadsides, culverts, and bridges) into farmlands using soil or stone bund water channels, small water storage ponds, terraces, and roadside pits [12]. These water channels are both advantageous and affordable. However, labor constraints, negative perceptions of road water harvesting practices, weak development agents contact, low levels of social capital, lack of security of land rights, low levels of household head education, and development practitioners lack awareness on such practices or the local context result in the low adoption rate of RWHP in the semi-arid regions [13,14].

Given the affordability of RWHP and the significant effect of erratic rainfall on crop yields, it seems that road water harvesting practice would be a good initiative in semi-arid areas [15]. For semi-arid areas, rain water harvesting in general, and water harvested from roads in particular, could improve infiltration of water in the soil profile [6], conserve nutrients from runoff [12,16,17], recharge groundwater potential [12], enhance moisture availability in plant roots, improve crop yields [15,18], and contribute to income and food security, at least for producers [8,9]. Above all, road water harvesting is considered a viable strategy for people living in semi-arid areas for countering droughts, mitigating flooding, and enhancing climate resilient agriculture by promoting use of inputs (such as fertilizer, compost, and improved seed), crop yields, and income [9]. In this regard, Hagos et al. [19] indicate that the use of water harvesting has resulted in an increase in per capita income of USD 82 per season and 24% less poverty among users of RWHP compared to non-users.

This paper aims to evaluate the impacts of RWHP on input uses (such as inorganic fertilizer and compost), crop yield, and income. In Ethiopia, there is limited evidence of factors affecting smallholders' uses of road water harvesting and its impacts on input uses, crop yields, and incomes. This paper presents an analysis of the determinants of the uses of road water harvesting and its socioeconomic impacts on subsistence farmers in the semi-arid areas of northern Ethiopia. The paper evaluates the demographic, socioeconomic, and biophysical factors affecting the use of road water harvesting and evaluates its impacts on input uses, yields, and income with particular reference to the Tigrai region of northern Ethiopia. The evidences generated from this paper would help policy makers in the dryland areas of the world particularly in sub-Saharan African countries to formulate

and promote appropriate road water harvesting strategies in the future, as the demand for water in the era of climate change is quite pertinent.

## 2. Materials and Methods

### 2.1. The Study Area

Ethiopia is geographically complex country that occupies broad agro-ecological zones that ranges from the lowest altitude at the Danakil depression at about 126 m below sea level, to up to the highest peak at 4562 m above sea level at Ras Dashen [20]. The temperature in Ethiopia ranges from 10 degree Celsius in the central Highlands to 40 degree Celsius in Danakil Depression that is considered the hottest place in the world. Following the geographical complexity, rainfall also varies (250 millimeters in Gode city to 2000 millimeters in the Central Highlands). Rainfall variability is the critical failure factor for food security in most parts of the country.

Tigrai is one among the nine Regional States of Ethiopia (see Figure 1). Tigrai Region is located in northern Ethiopia and belongs to the African dry land zones often called the Sudano–Sahalian region [21]. It is lies in northern Ethiopia, extending from 120 to 150 north latitude and 3630'' to 4130'' east longitude. The region is bounded to the north by Eritrea, to the west by the Sudan, to the south by the Amhara region, and to the east by the Afar region. Altitude varies from about 500 meters above sea level (m.a.s.l.) in the northeast to almost 4000 m.a.s.l. in the southwest [22]. The region is divided into six zones, which in turn are divided in to 98 districts. The population size of the region is estimated around six million, with an annual growth rate of 2.5 percent occupying an area of just over 80 thousand square kilometers. The average population density of the region is 80 persons/km2, with high concentrations in the Eastern, Southern, and Central Zones where it is 131, 122, and 115 persons/km2, respectively [23]. Agriculture and its allied activities constituted about 55% of the regional GDP and provided employment for more than 85% of the population. The farming systems of the region are largely based on traditional technologies and practices. The production system is characterized by scarcity of arable land, highly fragmented farm plots, and highly variable and insufficient rainfall [21].

Average annual rainfall in Tigrai is between 552 mm and 767 mm per year. The precipitation occurs mostly during a short summer (end of June to mid-September) rainy season, often falling as intense storms that result in serious soil erosion and yield reduction. However, since the 1980s, many areas of the natural resources have reappeared on hillsides following agreements by local communities to restrict access by people and grazing animals to these areas [5]. The land use pattern of the region indicates that out of the total land size of the region about 1.06 million hectares are suitable for cultivation [22]. The average land holding per household in the region varies from one location to another; in areas where the population density is very high, the average size of holdings ranges between 0.5–2.0 hectares, while in sparsely populated areas, like in western low land areas of the region, it goes up to 4 hectares or more [22].

The major crop production constraints in the region are: Poor agronomic and soil management practices coupled with the lack of improved farm implements; high moisture stress; recurrent outbreak and expansion of crop pests, diseases, weeds infestation and rodents, small size of land holding, shortage of oxen, severe soil erosion, and poor soil fertility [21]. Soil erosion, soil nutrient depletion, moisture stress, deforestation, and overgrazing are the major environmental problems in the region. In general, Tigrai contains many of the areas of greatest land degradation concern in Ethiopia's highlands. Erosion and the resulting land degradation have become serious problems in the region, while the use of agricultural inputs, such as improved seeds and fertilizer, is at a lower level. Moreover, farmers have poor access to essential supporting services. As a result, crop yields are very low. Recurrent drought and low yields have caused the growth rate of food production to lag behind the rate of population growth [22].

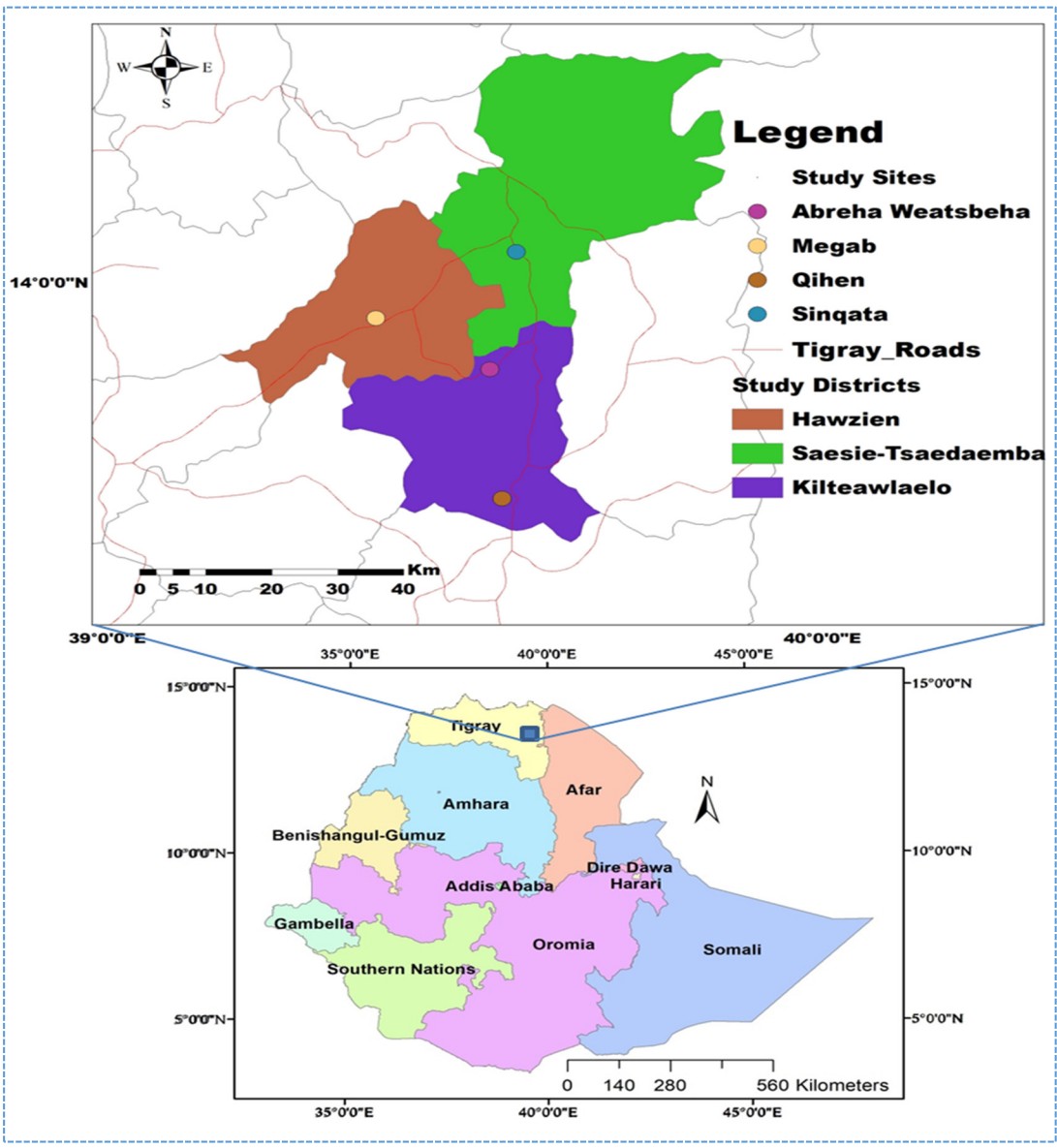

**Figure 1.** Location of the study area, Ethiopia.

Several factors make Tigrai suitable for road water harvesting. The first factor is the region's rainfall occurrence and distribution. The climate in Tigrai is characterized as semi-arid. Mean annual precipitation is variable, with erratic and torrential rains mainly (70%) concentrated in the period of June to early September, with considerable inter-year variability [20]. The inter-year variability of precipitation has two main consequences: Soil erosion during the rainy season, with Tigrai considered to be seriously degraded, and water insecurity during the dry season, with severe implications in terms of yield, income, and food security.

In both paved and unpaved roads, water harvesting practices have been implemented systematically in Tigrai since 2014 in an attempt to protect rural roads from rainfall damage, reduce landscape degradation, and promote water and food security [24]. Depending on several factors (including topography, soil types, rainfall amount, and water demand), different techniques of road water harvesting were implemented: (a) Channeling water from hydraulic systems (bridges, culverts, and road side drainages) into a series of deep trenches for enhancing soil moisture and groundwater recharge (e.g., Figure 2a,b), (b) channeling water from culverts and road sides into farm lands (e.g., Figure 2c), (c) use of ponds to harvest water from roads (culverts, bridges) for surface water storage

and groundwater recharge (e.g., Figure 2d), (d) shallow groundwater development upstream of Irish bridges and fords, and (e) spring capture in road cuts.

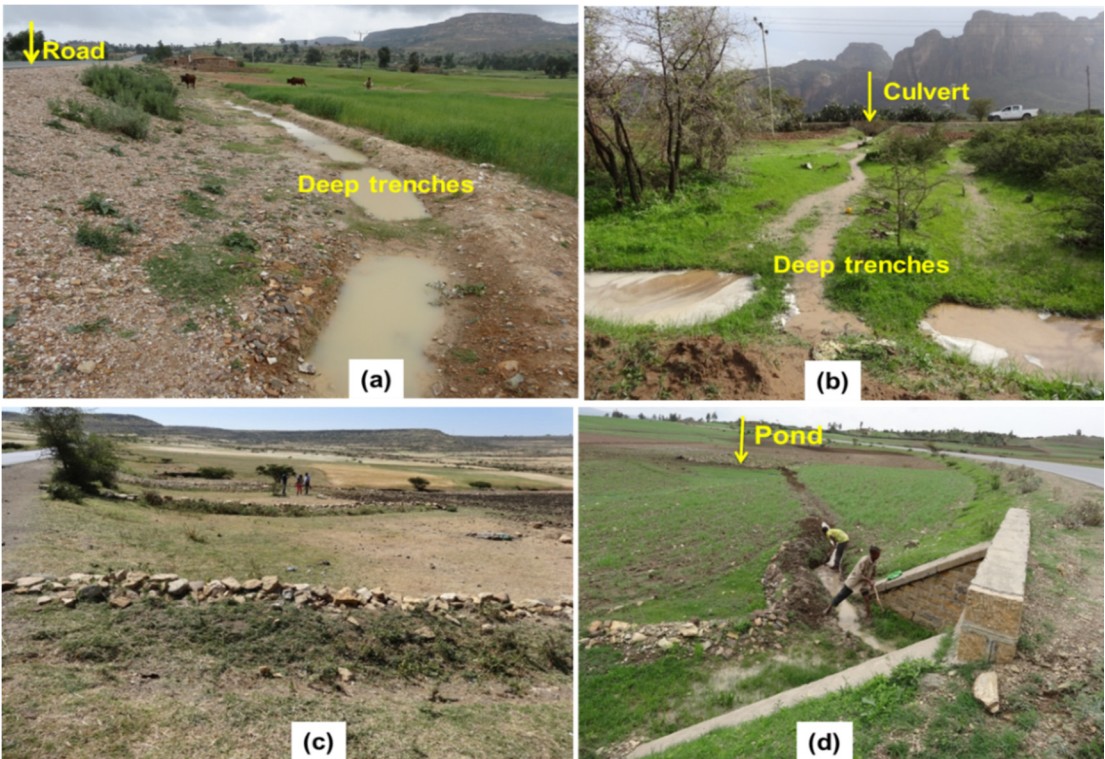

**Figure 2.** Examples of the different types of road water harvesting implemented in northern Ethiopia: Channeling water from roads into series of deep trenches for enhancing soil moisture and recharging groundwater (**a**,**b**); channeling water from roadside into farm lands (with no trenches) for enhancing soil moisture and productivity (**c**); and water from culvert is stored in ponds for surface water use and groundwater recharge (**d**). (Photo: Authors).

This road water harvesting practice has been part of a larger watershed development campaign throughout the region. Since the 1980s, soil and water conservation (SWC) and water harvesting techniques have been widely implemented to tackle land degradation and foster development. SWC and water harvesting reduce surface runoff and enhance infiltration, sediment deposition, and vegetation growth [25]. There are many options for joint SWC and road water harvesting initiatives, such as diverting water from culverts, using springs that are opened up with road construction, or reusing excavation pits as storage reservoirs. One such initiative was undertaken along the upgraded Qihin-Wukro-Freweign-Hawzien-Abreha Weatsbeha-Wukro route in Tigrai. This 64 km road section crosses three woredas (districts): Saesie Tsaeda Emba (woreda center is Freweign town), Hawzien woreda (woreda center is Hawzien town), and Klite Awlaelo woreda (woreda center is Wukro town), where this study has been conducted. This road rout has been selected for this study due to the ongoing efforts by the development practitioners in promoting road water harvesting and the availability of long-term hydrological monitoring of the interventions in the area. Both paved and unpaved roads were considered in this study.

## 2.2. Sampling Procedures and Data

Multi-stage sampling procedures were employed in order to obtain representative information. First, a list of villages and their road water harvesting status were identified via field visits, and then households were randomly selected from the list of each kebele's (in Ethiopia, kebele is the lowest administrative unit) farmers. Survey data were collected through home visits between March and April

2018, through which four trained interviewers, under the supervision of the principal researcher, spoke to 159 farmers of 603 plots. The household-level interviews were conducted during home visits using semi-structure interview schedule. The major data set for the interview schedule includes household demographic, socio-economics, land use, use of road water harvesting, yield, input use, income sources, and farmers' viewpoints concerning the effects of road water harvesting and the perceived benefits and losses related to the road water harvesting practices.

*2.3. Analytical Procedures*

A binary probit regression model was used to analyze factors affecting the use of water harvesting. Binary choice models assume that individuals are faced with a choice between two alternatives and that their choice depends on the characteristics of the choices. The Propensity Score Matching (PSM) technique was used to analyze the impact of participation on household yield, productivity, and income status. The propensity score is defined as the probability of receiving an intervention given pre-treatment covariates [26]. Evaluating a program or an intervention is always a problem when there is no baseline survey and when the subjects receiving the treatment are assigned in a non-random manner. To reduce bias in the estimation of treatment effects, the technique of matching subjects that received treatment (use of water harvesting) with those that did not receive treatment (non-users of water harvesting) based on observable covariates was developed [27].

Matching based on propensity scores is a mechanism developed to control bias in the estimation of treatment effects (impact) that arise due to confounding factors. Once the propensity scores are established, the average treatment effects of water harvesting on the water harvesting practices of adopters households or average treatment effect on treated (ATT)—in this study, the amount of farm yard manure used, amount fertilizer used, crop yield, and annual crop income status matrixes—are estimated using different matching algorithms [27] that use different techniques to match the two comparison groups based on their propensity scores (for details of model definition, see Appendix A). In this study, all the outcome variables have been estimated at plot level.

The matching algorithms used in this study were Nearest Neighbor Matching (NN), Radius Matching (RR), Kernel Matching (KK), and Stratification Matching (SS). These algorithms differ in the way 'neighbor' is defined for each treated individual, how the common support is handled, and how the weights are assigned to these neighbors. Nearest Neighbor, Radius, Kernel Matching, and Stratification Matching are algorithms used to measure the impacts of development intervention following their different inherent attributes. For instance, the Stratification method deals with dividing the range of variation of the propensity score in intervals such that within each interval treated and control units have on average the same propensity score. The ATT of interest is finally obtained as an average of the ATT of each block with weights given by the distribution of treated units across blocks. The Stratification method discards observations in blocks where either treated or control units are absent, resulting in a dropping number of observations. This observation suggests an alternative way to match treated and control units, which consists of taking each treated unit and searching for the control unit with the closest propensity score, i.e., the Nearest Neighbor. Once each treated unit is matched with a control unit, the difference between the outcome of the treated units and the outcome of the matched control units is computed. The ATT of interest is then obtained by averaging these differences. While in the case of the Stratification method, there may be treated units which are discarded because no control is available in the block, in the case of the Nearest Neighbor method, all treated units find a match. The Radius Matching and Kernel Matching methods offer a solution to this problem. With Radius Matching, each treated unit is matched only with the control units whose propensity scores fall in a predefined neighborhood of the propensity score of the treated unit. If the dimension of the neighborhood is set to be very small, it is possible that some treated units are not matched because the neighborhood does not contain control units. With Kernel Matching, all treated are matched with a weighted average of all controls with weights that are inversely proportional to the distance between the propensity scores of the treated and controls. It is clear from the above considerations

that these four methods reach different points on the frontier of the trade-off between quality and quantity of the matches, and none of them is a priori superior to the others. Hence, the variation in the number of observations during matching by the four algorithms is normal and acceptable. Their joint consideration, however, offers a way to assess the robustness of the estimates [27].

For detailed discussions about the Propensity Score Matching (PSM) model specification, assumptions, and matching algorithms, see [24]. The outputs of the PSM follow all the necessary steps and fulfill the assumptions of the PSM models and algorithms. The estimates confirm a significant reduction in bias from the matching procedure, indicating the equality of characteristics across the user and non-user groups [28]. To further control for selection bias associated with unobserved characteristics, we also estimate the sensitivity of our results to the potential hidden bias from these confounding factors. We also perform a variety of robustness checks that increase the confidence of our treatment effects [28]. To identify explanatory variables, we draw on literatures (See Table 1) that emphasizes the importance of productive resources and social capitals as determinants of technology adoption decisions [7,11–14,19–23]. Quantitative data analyses were carried out using Stata software version 14.

**Table 1.** The synthesis of the study variables and measurement indicators.

| Definition of Variable | | Measurement |
|---|---|---|
| **Confounding Factors** | **Household Characteristics** | |
| | Age of household head | Year |
| | Family size | Number |
| | Dependency ratio | The ratio of dependent and active family |
| | Livestock size | Size of livestock own in Tropical livestock unit (TLU) |
| | Frequency of development agent contact | Number of days in contact with development agent per year |
| | Sex of household head | 1 = male, 0 = female |
| | Literacy status of household head | 1 = literate, 0 otherwise |
| | Education level of household head | Years of schooling |
| | Access to credit | 1 = yes, 0 = no |
| | Access to rainwater harvesting training | 1 if the household has access to rainwater harvesting training, 0 otherwise |
| | Distance from home to district market | Minute |
| | Number of plots owned | Number |
| | **Plot Characteristics** | |
| | Use of improved seed | 1 = yes, 0 = no |
| | Plot size | Timad (one Timad is 0.25 hectare) |
| | Plot distance from dwelling | Minute |
| | Land tenure status | 1 if the plot is owned, 0 if the plot is not owned |
| | Plot distance from road | Minute |

**Table 1.** *Cont.*

| Definition of Variable | | Measurement |
| --- | --- | --- |
| **Outcome Variables** | Participation status in road water harvesting | 1 if the household participates with the plot in the use of RWHP, 0 otherwise |
| | Crop yield | Total amount of kilograms of harvest per plot |
| | Crop productivity | Amount of yield per timad |
| | Annual crop income | Total income obtained per plot from crop sell in Ethiopian Birr (1 Ethiopian Birr (ETB) is equivalent to 0.027 US dollar (NBE, 10 October 2020) |
| | Amount of fertilizer used | Amount of kilograms of inorganic fertilizer used per plot |
| | Amount of farmyard manure used | Amount of kilograms of farm yard manure used per plot |

Previous authors have suggested the need for a normality test in statistical analysis of numeric dependent variables. Normality can be tested either by graphical or numerical approaches. The numeric approaches are useful for making objective judgments of normality, but less sensitive for small sample size or overly sensitive for large sample sizes, in this case graphical test of normality is more preferable [29]. There are numerous graphic methods to test the normality continuous data. The well-known graphic normality tests includes P–P Plot (widely known as probability–probability plot or standardized plots), box plot, Q–Q Plot (quantile–quantile plot), Shapiro–Wilk test, Kolmogorov–Smirnov test, and histograms [30]. As compared to other graphic normality tests, P–P plots are more precise for large data sets. A P–P plot is a probability plot for assessing how closely fit the expected and observed value of a given data sets. When the data sets are normality distributed, the P–P plots become approximately a straight line. Data far apart from this straight line indicates the existence of outliers and a lack of normality in the data set. By visualizing the P–P plot, one can make a decision about outliers, skewness, and kurtosis, and hence this method of normality test has become a very popular tool for testing the normality assumption [31]. Following these extra advantages, its practical simplicity, and strengths in applied research, the P–P plot normality test was used in this paper (see Appendix C).

## 3. Results and Discussion

### 3.1. Descriptive Results

Table 2 presents the profiles of the sample households. The descriptive results indicate that non-users of road water harvesting were older (51.54 years) than the heads of user households (50.80 years). Households that engage in road water harvesting had fewer economically dependent family members (1.27) than non-user households (1.51). RWH-user households owned more productive resources (e.g., land and livestock) (4.88) than non-user households (4.48). The average plot size for both users and non-users was 1.27 and 1.16 tsimd, respectively, implying slightly larger plot sizes for RWH households. On average, users and non-users owned 2.7 and 2.69 plots, respectively.

On average, development agents contacted users 3.94 times per year and non-users 4 times per year. In relation to education level, 68.63% and 32.13% of users and non-users, respectively, were formally educated. This implies that a higher percentage of households with literate household heads were beneficiaries of road water harvesting practices.

**Table 2.** Profiles of the sample household (*n* = 159) and plots (*n* = 603).

| Household Characteristics | Variable Means by Road Water Harvesting Practice (Mean/Percent) | |
|---|---|---|
| | **Non-User** | **User** |
| Age of household head (year) | 51.54 | 50.80 |
| Family size (number) | 6.28 | 5.67 |
| Dependency ratio (number) | 1.51 | 1.27 |
| Livestock size (TLU) | 4.48 | 4.88 |
| Frequency of development agents contact (number of contacts per year) | 3.94 | 4.0 |
| Sex of household head<br>Female (%)<br>Male (%) | <br>54.74<br>67.13 | <br>45.26<br>32.13 |
| Literacy status of household head (%) | 32.13 | 68.63 |
| Education level of household head (years of schooling) | 1.38 | 2.56 |
| Access to credit (%) | 31.86 | 32.08 |
| Access to rainwater harvesting training (%) | 31.94 | 32.16 |
| Distance from home to district market (minutes) | 62.75 | 49.76 |
| Number of plots owned (number) | 2.69 | 2.73 |
| **Plot Characteristics** | **Non-User** | **User** |
| Use of improved seed (%) | 54.2 | 48.8 |
| Plot size (timad) | 1.16 | 1.27 |
| Plot distance from dwelling (minutes) | 25.00. | 16.03 |
| Land tenure status (%) | 16.3 | 83.7 |
| Plot distance from road (minutes) | 10.80 | 2.01 |
| Crop yield (kg/plot) | 240.08 | 317.56 |
| Crop productivity (kg/timad) | 805.93 | 904.89 |
| Annual crop income (Ethiopian Birr/plot) | 5432.58 | 7811.22 |
| Amount of fertilizer used (kg/plot) | 27.03 | 37.53 |
| Amount of farmyard manure used (kg/plot) | 137.51 | 296.80 |

The amount of fertilizer used by users and non-users was 37.53 kg/ha and 27.03 kg/ha, respectively. Similarly, use of farmyard manure was higher for user households (296.80 kg/ha) compared to non-user households (137.56 kg/ha). Road water harvesting users apply more inputs (e.g., fertilizer and farmyard manure) compared to non-users. Users travel about 50 minutes to reach market centers, while non-users reported travelling 73 minutes. On average, non-user plots were about a 10-minute walk from the road and a 25-minute walk from their home. On the other hand, user plots were a 2-minute walk from the road and a 16-minute walk from home.

As can be seen from the Table 3, from the 159 sample households 26 (16.4%) of them were female headed households, while 133 (83.6%) of them were male headed households (see Table 3). Of the total of female headed households, 45.26% of them were water harvesting users, while from the total of sample male headed households, only 32.13% of them were users of road water harvesting (see Table 3). Though the size of female sample household is smaller than male headed households, the intra gender household (male/female) analysis result indicates the presences of higher proportion of females in relation to uses of road water harvesting compared to that of the intra male gender headed households.

**Table 3.** Gender of the household heads (*n* = 159).

| Sex of the Household Head | Frequency | Percent |
|:---:|:---:|:---:|
| Female | 26 | 16.4 |
| Male | 133 | 83.6 |
| Total | 159 | 100.0 |

*3.2. Factors Affecting Adoption of Road Water Harvesting Practices*

Various statistical methods used for data analysis make assumptions about normality, including correlation, regression, t-tests, and analysis of variance. If a continuous data follows normal distribution, then it is recommended to present such data in mean values. These values are further used to compare between the groups by calculating the significance level. If data are not normally distributed, the resultant mean will not be a representative value of the data set. A wrong selection of the representative value of data set and further calculated significance level using this representative value might give wrong interpretation [29]. Following this background, in this paper, first we test the normality of the data, then we checked whether the mean is applicable as a representative value of the data or not. Once the mean is found applicable, the RWHP users and non-users group mean was compared using parametric test, otherwise, medians would have been used to compare the groups, using nonparametric methods. Such an approach has been widely used by several authors in impact evaluation researches [27,32–34].

To examine factors that predict a farmer's adoption or non-adoption of RWHP, we use a binary probit regression model in which adoption takes the value of 1 and non-adoption takes the value of 0 as a dummy dependent variable at a particular time (i.e., during the survey year). Table 4 presents the factors that affect the adoption of road water harvesting practices. Of the 11 variables that are thought to influence adoption or non-adoption of RWHP, seven are found to be a statistically significant influence on a farmer's probability of adoption. The regression result indicates the presence of enough information in the model to explain factors affecting uses/non-uses of water harvesting ($x^2$ = 47.20, $p$ = 0.00). The results suggest that both socioeconomic and plot characteristics are statistically significant in conditioning a household's decision to adopt road water harvesting practices. The results further suggest that determinants of adoption can be broadly classified into social characteristics of the household head, labor availability, and plot characteristics, which includes plot size and plot distance from the infrastructure.

Increased distance from home to district market negatively and statistically significant influences the adoption of RWH. This variable was statistically significant with $p < 0.01$ in the model. The marginal effect of this variable indicates that as the walking distance to the district market center from the farmer's residence increases by one minute from the mean value, the probability of adopting RWHP decreases by 0.1 percent. The literature also shows a decrease in the rate of technology adoption with an increase in the distance from markets [6]. Proximity is therefore an important factor in a farmer's decision to adopt technology, as it facilitates the spatial integration of the product and factor markets. Better market connections reduce the transaction costs caused by information asymmetry and increase the availability and uses of the support services that can promote technology adoption [34]. An empirical study in Ethiopia found that access to markets and a shorter distance from rural towns affects the purchase of inputs by rural households [19,24]. The negative relationship indicates that households living far from roads are less likely to use road water harvesting practices; as such, plots get less attention from these households and are less accessible via the small structures constructed through road water harvesting projects. Prior to the systematic implementation of road water harvesting, empirical evidence [35] shows that plots close to roads are negatively affected by water from roads (flooding, siltation, erosion). After the implementation of road water harvesting, plots close to roads benefit more than those far from roads for two main reasons: Farmers are able to connect road hydraulic structures to their farms

and associated water storage units, and the available water from roads is limited, so plots far from roads receive less runoff.

**Table 4.** Binary probit model on the adoption of road water harvesting technology (*n* = 603).

| Variable Characteristics | Variables | Coefficients | Marginal Effects | Z-Value |
|---|---|---|---|---|
| **Household Characteristics** | Age of household head | −0.04 | −0.14 | −0.64 |
| | Dependency ratio | −0.24 | −0.08 | −3.56 *** |
| | Livestock size | 0.02 | 0.007 | 1.05 |
| | Frequency of development agent contact | 0.02 | 0.006 | 1.10 |
| | Education level of household head | 0.48 | 0.17 | 3.41 *** |
| | Access to credit | −0.15 | −0.05 | −1.24 |
| | Distance from home to district market | −0.001 | −0.001 | −1.66 * |
| **Plot Characteristics** | Use of improved seed | 0.52 | 0.02 | 4.23 *** |
| | Plot size | 0.07 | 0.02 | 2.17 ** |
| | Plot distance from dwelling | −0.01 | −0.003 | −2.91 *** |
| | Land tenure status | 0.28 | 0.09 | 1.83 * |
| **Model summery** | Constant | −0.44 | | −0.84 |
| | Number of observations | 603 | | |
| | log likelihood | −343.66 | | |
| | LR X$^2$(5) | 68.74 | | |
| | Prob > X$^2$ | 0.000 | | |

Note: *, **, and *** represent statistical significances at the 10%, 5%, and 1% levels, respectively.

Similarly, increased distance between the farmer's dwelling and the farm plot has a negative effect on the adoption of road water harvesting. The results suggest that as the distance between the plot and the farmer's residence increases by one minute (walking), the probability of the use of RWHP decreases by 0.3%. This is believed to be because the closer the plot is to the dwelling, the easier it is for the family to supervise and manage the farm [14,21].

In addition, we found that an increase in land size has a statistically significant and positive impact on the adoption of RWHP. Land size was statistically significant, with less than a 5% probability level in the model. The marginal effect results indicate that, as the size of the plot increases by one hectare from the mean, the probability of the use of RWH increases by 2%. The positive association between the probability of adopting RWHP and farm size suggests the presence of economies of scale [21,22]. Moreover, farm size is often used as an indicator of wealth in agrarian settings, and the results here may suggest that wealthier households are more likely to adopt innovations, because they may be more able and willing to bear risks than their counterparts, and they may have preferential access to inputs and credit.

Schooling is crucial to creating awareness and attitudes towards technology adoption. In this study, years of schooling is found to positively influence the adoption of RWH technology. Our results show that as years of schooling increase by one year, the probability of RWHP use increases by 17%; this is similar to the findings of [24,36]. This is because education increases access to information about innovation, inputs and their uses, and the management of technology, all of which are crucial to creating positive attitudes towards technology adoption. Increasing farmers' education via adult learning and tailor-made training can thus be used as a tool to encourage the adoption of specific agricultural practices.

Increasing the non-economically active family size of the household, as indicated by the dependency ratio, is found to be negative and statistically significant at less than 1% probability for the use of RWHP. Keeping other factors constant, the marginal effect indicated that as the dependency ratio increased by one, the probability of using road water harvesting decreased by 8% from the mean.

Labor issues seem to be of more concern in the decision to adopt RWHP. Specifically, the probability of the use of RWHP increased with increased family size: An increase in family size results in an increase in the number of household members who actively provided farm labor. This underscores the importance of labor availability to the adoption of labor-intensive technologies [17]. In such circumstances, it is important to consider strengthening and structuring existing local labor-sharing mechanisms [21]. On the other hand, the probability of implementing RWHT declined with the number of dependents in the household, capturing the intuitive expectation that the time spent caring for dependents shifts labor away from labor-intensive activities.

The results of this survey revealed that the use of improved seed is an important variable in differentiating adopters from non-adopters of RWHP in the study area. The coefficient of this variable is positive and statistically significant at less than 1 percent probability level. Keeping all other factors constant, the marginal effect of using improved seed results in a 2% increase in the probability of adopting RWHP. Based on this result, we can conclude that experience in any extension program can enable farmers to widen their knowledge of modern farm operations, including the adoption of RWHP. Most importantly, experience in such programs enables farmers to perceive the risks and relative advantages of new agricultural technologies more accurately.

With regard to land ownership, as indicated by land tenure status, our results show that the probability of adopting RWHP is positively and statistically significant associated with farmers' ownership of their plots, implying that rented plots are less likely to be selected for RWHP use compared to owner operated plots. Farmers use RWHP on their own plots, rather than on leased plots; tenure arrangements hinder RWHP adoption. Investment in RWHP decisions are related to ownership security; the longer a farmer has owned their plot, the more likely it is that they will engage in land management practices [37].

### 3.3. Impacts of Road Water Harvesting

The propensity score matching technique was used to analyze the impact of participation in RWHP on uses of farmyard manure and fertilizer, yield, and income. The propensity score is defined as the probability of receiving an intervention given pre-treatment covariates. In our analysis on the impact of RWHP, we compared plots that used RWHP and those that did not (control) in terms of uses of farm yard manure and fertilizer use, yield (physical grain yield (kg/plot) of the respective crops), and income obtained from the major crops grown in the area (cereal and bean).

The performance of the matching model was checked through different tests. For this, the common support region [0.11463415, 0.9895831], which ensures that the mean propensity scores for RWHP users and non-users was selected. The common support region is a region where the values of propensity scores of both adopters and comparison groups were defined. The region of common support will be defined by dropping observations below the maximum of the minimums and above the minimum of the maximums of the balancing scores between the two groups. Then, the Average Treatment Effect on treated (ATT) are only determined in the region of common support. Average treated effect on treated or ATT is the impact of adoption of the RWHP on the adopters in comparison with the non-adopters.

The balancing property is satisfied when the number of blocks is five. In addition, we ran a paired t-test analysis on the covariates used to match adopter with non-adopter households. The difference between these households was statistically insignificant after matching (for details of the robustness of the PSM See Appendix B).

### 3.3.1. Impact of the Adoption of Road Water Harvesting Practices on Farmyard Manure and Fertilizer Use

As shown in Table 5, RWHP users applied a higher rate of fertilizer compared to non-users. Results from NN and SS show that users applied 5.06 ($p < 1\%$) up to 8.11 kg ($p < 5\%$) extra inorganic fertilizer compared to non-users. Similarly, RWHP users were also found to apply additional farmyard manure of 119 kg to 149 kg. This implies that the adoption of road water harvesting has a positive

impact on farmyard manure consumption. These results show that the use of RWHP is important in promoting complementary land management practices in the study area, as the coefficient of both farmyard manure and fertilizer is positive and statistically significant; input uses are said to be more responsive to water that results in higher production. Based on this result, we can conclude that experience in any extension program can enable farmers to widen their knowledge of modern farm operations, including the adoption of fertilizer. Most importantly, farmers know that moisture is the major constraint in the application of inputs (fertilizers, etc.), and that ensuring adequate moisture reduces the risk of crop failure.

**Table 5.** Impacts of road water harvesting, n = 603.

| Outcomes | Algorithms | Number of Plots of RWHP Users | Number of Plots of RWHP Non-Users | Average Treatment Effect on Treated (ATT/impact) | Std. Err | T-Value |
|---|---|---|---|---|---|---|
| Amount of fertilizer used | Nearest neighbor matching (NN) | 193 | 138 | 8.109 | 3.457 | 2.35 ** |
| | Radius matching (RR) | 193 | 404 | 9.60 | 2.27 | 3.58 |
| | Kernel matching (KK) | 193 | 404 | 5.06 | 3.84 | 1.46 |
| | Stratification matching (SS) | 193 | 404 | 5.36 | 0.84 | 6.35 *** |
| Amount of farm yard manure used | Nearest neighbor matching (NN) | 193 | 138 | 139.84 | 71.45 | 1.96 * |
| | Radius matching (RR) | 193 | 404 | 148.60 | 76.83 | 1.93 * |
| | Kernel matching (KK) | 193 | 404 | 130.28 | 68.31 | 1.91 * |
| | Stratification matching (SS) | 193 | 404 | 119.21 | 61.42 | 1.94 * |
| Crop yield | Nearest neighbor matching (NN) | 193 | 138 | 106.19 | 36.11 | 2.94 *** |
| | Radius matching (RR) | 193 | 138 | 71.92 | 22.73 | 3.21 *** |
| | Kernel matching (KK) | 193 | 404 | 59.84 | 14.84 | 4.03 *** |
| | Stratification matching (SS) | 193 | 404 | 59.56 | 25.27 | 2.36 ** |
| Annual crop income | Nearest neighbor matching (NN) | 193 | 138 | 3085.81 | 1543.42 | 2.00 ** |
| | Radius matching (RR) | 193 | 404 | 2351.56 | 900.79 | 2.61 *** |
| | Kernel matching (KK) | 193 | 404 | 2512.72 | 3337.18 | 0.75 |
| | Stratification matching (SS) | 193 | 404 | 2556.04 | 1.351.20 | 1.89 * |

Note: *, **, and *** represent the statistical significances at the 10%, 5%, and 1% levels, respectively.

### 3.3.2. Impact of the Adoption of Road Water Harvesting Practices on Crop Yield

The average computed difference on crop yield between RWHP user and non-user households showed a statistically significant variation. The PSM results presented in Table 4 indicate that the RWHP user households, on average, gained additional yields of 60 kg to106 kg per plot compared to non-user households. The result is significant at $p < 0.001$ for NN, KK, RR matching, and at $p < 0.05$ for SS matching. These results imply that compared to control plots, plots with RWHP gain statistically significant higher yield levels. Specifically, RWHP positively and statistically significantly impacts yield, not only through increasing water availability, but also by decreasing weeds and facilitating

crop growth, enabling plots to escape moisture stress during critical crop growth stages in dryland areas. In addition, RWHP also increases farmers' confidence to invest in the crop through yield enhancing inputs such as inorganic fertilizer and composites. This is in line with the descriptive statistics results (see Table 2), where the amount of fertilizer and farmyard manure used is reported as higher for RWHP-user households compared to that of non-users, indicating that RWHP use promotes agricultural intensification through better input use and management practices. Water resilience in agriculture aims at safeguarding water availability during periods of shocks, such as persistent droughts [15]. There is hence a clear link between making use of roads for water and increased productivity and resilience [6,7,14]. Some authors [38] estimate that every 10% increase in yields in Africa leads to a 7% reduction in poverty.

### 3.3.3. Impact of the Adoption of Road Water Harvesting Practices on Household Income

The average computed difference in income between RWHP user and non-user households showed a statistically significant variation. The PSM results presented in Table 5 indicate that RWHP user households, on average, gain additional yields of 2512 to 3085 Ethiopian Birrs compared to non-user households. The result is statistically significant at $p < 0.001$ for RR, at $p < 0.05$ for NN, and at $p < 0.1$ for SS matching. This implies that there is a statistically significant mean income difference between households that adopt RWHP and non-adopters. This finding is consistent with the findings of various authors [4,36] who report that RWHP has a great impact on farm income.

### 4. Conclusions and Policy Implications

In this paper, we used both household and plot-level data from the semi-arid region of Tigrai, Ethiopia to investigate the factors influencing farmers' decisions to adopt RWH practices and its impacts on input use, yield, and farm income. Binary probit regression and PSM were used to analyze the data. With regard to factors that influence use of RWH, our results underscore the importance of both plot and household characteristics on adoption decisions. Our findings imply that public policy can affect the promotion of RWHP in drylands if based on an understanding of both household and farm characteristics. The casual impact estimation from the propensity score matching suggests that RWHP has positive and significant impacts on input uses (farmyard manure and fertilizer), crop yield, and farm income among the sample households.

The findings presented in this paper have far-reaching implications for emerging climate-induced agricultural challenges. Although relatively forgotten and underutilized, capturing water from roadside drains, culverts, or road embankments is, in many cases, the easiest way to capture runoff. The network of roads is fine-grained and, in many areas, fast increasing. The ability to better retain water will help farmers to tide over drought periods and increase their capacity to deal with shocks. Results from this research show that supplementary irrigation with water from roads increases input use, crop yields, and farm income by mitigating intra-seasonal dry spells in the month of September, which is the crop maturity period. Moreover, implementing water harvesting systems reduces the risk of crop failure, making farmers more willing to invest in fertilizers and other agricultural inputs [19], which will further increase crop yields and enhance resilience to climate-related shocks [39]. In this paper, we argue that road water harvesting provides opportunities for packages of technology adoption, such as yield enhancing inputs and a strategy to build adaptive capacity against shocks and extreme events, by providing an extra source of water during dry spells, increasing soil moisture, and reducing the risk of floods. In addition, water can be stored in ponds, shallow wells, and small dams, and can be used for livestock or a second round of cash crop production during the dry season. This will provide extra sources of income and therefore increase farmers' resilience against adversities. Therefore, the results suggest that policy interventions should encourage development of multifunctional physical infrastructures (such as road) and promotion of community-based conservation behavior via the well-known grassroot-oriented informal social networks mechanism among others.

Some of the limitations of this study should be outlined. First, the data for this study were limited to one point in time; as such, there was no temporal component to the analysis. Second, this study only used a quantitative approach, and as such, some non-quantifiable qualitative variables that affect uses of RWHP and its impacts may not be fully captured. Hence, while our goal was to show the causal relationship between road water harvesting and its impacts, we recommend the use of panel and mixed method approaches as an important next step to guide future research on the relationships between RWHP and its multidimensional impacts.

**Author Contributions:** As principal author, K.M.G. designed this research, collected the data, analyzed the result, and wrote the draft article. A.B. and K.T.G. participated in data collection, data analysis, and literature review and drafting the manuscript. K.W.; F.v.S.; L.F.V., and T.A. participated in supporting the design, framing the overall data analysis, interpreting the results, and reviewing the manuscript. All authors have read and agreed to the published version of the manuscript.

**Funding:** This research received no external funding.

**Acknowledgments:** The support of the Green Future Farming program and Global Resilience in the preparation of this paper is acknowledged.

**Conflicts of Interest:** The authors declare that they have no conflict of interest.

## Appendix A. Econometric Model Estimation Procedures

The determinant of factors affecting the adoption of water harvesting technology was analyzed using a binary probit model, which takes the value of 1 for road water harvesting users and 0 for non-users. The probit model was employed because it is useful when an individual must choose between two alternatives (in this case, either to adopt RWHP or not). A binary probit regression is used to regress the dependent variable, Y, of whether the farmer had adopted RWHP:

Prob (event) = Prob (Y, 1 represents ith farmer adopted, and 0, otherwise)

$$
\begin{aligned}
Y &= \quad 1: \textit{adopted} \\
Y &= \quad 0: \textit{otherwise}
\end{aligned}
\tag{A1}
$$

$$
Y = Xi\beta i + u
\tag{A2}
$$

The probit model relates the probability of occurrence P of the outcome counted by Y to the predictor variables X. The model takes the form

$$
P(X) = \Phi(\beta 0 + \beta 1\, X1 + \beta 2\, X2 + + \beta k\, Xk)
\tag{A3}
$$

where $\Phi(Z)$ is the standard normal cumulative distribution function.

The probit model stands for the cumulative normal probability function as below.

$$
Y = \beta 0 + \beta 1\,(X1) + \beta 2\,(X2) + \dots \dots \dots + \beta n\,(Xn) + \varepsilon i
\tag{A4}
$$

where: Y is the probability of the farmer's participation in RWH; B is the parameters that are estimated by the maximum likelihood; Xi is a vector of exogenous variables that explain participation in RWH; and $\varepsilon i$ is the error term.

The effect of the adoption of water harvesting technology on input use, yield, and household income is analyzed using the propensity score matching (PSM) model. PSM gives information by comparing how the outcome variable differs for adopters and non-adopters of RWHP. The study examines the effect of the adoption of RWHP on crop productivity, yield, and household income. PSM enables us to see these effects, because it collects cross-sectional data. The probability of receiving a treatment given control characteristics is:

$$
P(X) = Pr\{D = 1|X\} = E\{D|X\}
\tag{A5}
$$

where D = {0, 1} is the indicator of exposure to treatment and X is the multidimensional vector of pre-treatment characteristics. PSM can be estimated using a probit or logistic model. The study employs the following binary probit model, because the data is cross-sectional:

$$yi = \beta 0 + \sum_{i=1}^{n} \beta iXi + \theta Di + \varepsilon i$$
$$n = 1, 2, 3$$

(A6)

where Yi = the dependent variable; Xi = vector of exogenous variable; βi = coefficient of the parameter.

Di = is whether the adopters of road water harvesting practices (Di = 1) or not (Di = 0).

To estimate the mean impact of the adoption of RWHP on crop productivity, yield, and household income is obtained by averaging the impact across all the individuals in the population. As a result, given a population of units denoted by i, if the propensity score p (Xi) is known, the Average Effect of Treatment on the Treated (ATT) can be estimated as follows:

$$ATT \equiv E \{Y1i - Y0i \,|Di = 1\}$$
$$ATT = E \{E \{Y1i - Y0i \,|Di = 1, p \,(Xi)\}\}$$
$$ATT = E \{E \{Y1i \,|Di = 1, p \,(Xi)\} - E \{Y0i \,|Di = 0, p \,(Xi)\} \,|Di = 1\}$$

(A7)

The estimate of the propensity score using a probit or logit model is not enough to estimate the ATT. The probability of observing two units with exactly the same propensity score value is, in principle, zero, because p(X) is a continuous variable. Therefore, this study used different methods of matching to overcome this problem, such as Nearest Neighbor Matching, Radius Matching, Kernel Matching, and Stratification Matching. The selection of these methods depended on the low mean and media bias, low pseudo R2, and the insignificance of variables after matching.

**Appendix B. PSM Model Fitness**

- Testing the balance of covariates

After chosen the best performing matching algorithm, it is important check the balancing of covariate using different procedures by applying the selected matching algorithm (nearest neighbor). The treated and control group have no significant difference regarding those variables after matching, because after matching all variables are insignificant. Clearly, after matching, the differences are no longer statistically significant, suggesting that matching helps reduce the bias associated with observable characteristics.

**Table A1.** The balance of covariates.

| Variable Characteristics | Variable | Before Matching (Mean) | | | After Matching (Mean) | | |
|---|---|---|---|---|---|---|---|
| | | Treated | Control | t-Test | Treated | Control | t-Test |
| Household Characteristics | Age of household head | 7.0471 | 7.1256 | 0.341 | 7.0563 | 7.068 | 0.912 |
| | Dependency ratio | 1.2723 | 1.5125 | 0.003 | 1.2899 | 1.2201 | 0.412 |
| | Livestock size | 4.8068 | 4.4806 | 0.206 | 4.5864 | 4.2425 | 0.203 |
| | Frequency of development agents contact | 3.9896 | 3.9366 | 0.857 | 4.0437 | 4.5355 | 0.183 |
| | Education level of household head | 0.40625 | 0.27317 | 0.001 | 0.38251 | 0.40437 | 0.670 |
| | Access to credit | 0.66667 | 0.66098 | 0.891 | 0.6776 | 0.62842 | 0.324 |
| | Distance from home to district market | 111.48 | 133.73 | 0.002 | 111.64 | 111.07 | 0.945 |

**Table A1.** *Cont.*

| Variable Characteristics | Variable | Before Matching (Mean) | | | After Matching (Mean) | | |
|---|---|---|---|---|---|---|---|
| | | Treated | Control | t-Test | Treated | Control | t-Test |
| **Plot Characteristics** | Use of improved seed | 0.39583 | 0.22195 | 0.000 | 39344 | 0.25683 | 0.005 |
| | Plot size | 4.3815 | 4.5244 | 0.488 | 4.2964 | 4.2787 | 0.940 |
| | Plot distance from dwelling | 15.417 | 24.81 | 0.000 | 15.787 | 16.885 | 0.637 |
| | Land tenure status | 0.84896 | 0.83171 | 0.594 | 0.85246 | 0.79235 | 0.13 |

- Sensitivity analysis

The last step in propensity score matching is conducting a sensitivity analysis to check unmeasured hidden variables. The rebound package was used to test the sensitivity of outcome variables. Table A2 indicated that the positive impact of outcomes is insensitive to unobserved selection bias.

**Table A2.** Sensitivity analysis.

| Outcome Variables | Gamma | sig+ | sig- | t-hat+ | t-hat- | CI+ | CI- |
|---|---|---|---|---|---|---|---|
| **Amount of fertilizer used** | 1 | 0 | 0 | 25 | 25 | 25 | 26 |
| | 1.05 | 0 | 0 | 25 | 25 | 25 | 27.5 |
| | 1.1 | 0 | 0 | 25 | 25 | 25 | 30 |
| | 1.15 | 0 | 0 | 25 | 25 | 25 | 30 |
| | 1.2 | 0 | 0 | 25 | 25 | 25 | 32.5 |
| | 1.25 | 0 | 0 | 25 | 27.5 | 25 | 32.5 |
| | 1.3 | 0 | 0 | 25 | 30 | 25 | 35 |
| | 1.35 | 0 | 0 | 25 | 30 | 25 | 37.5 |
| | 1.4 | 0 | 0 | 25 | 30 | 25 | 37.5 |
| | 1.45 | 0 | 0 | 25 | 32.5 | 25 | 37.5 |
| | 1.5 | 0 | 0 | 25 | 32.5 | 25 | 37.5 |
| | 1 | 0 | 0 | 25 | 25 | 25 | 26 |
| **Amount of Farmyard Manure Used** | 1 | 0 | 0 | 3 | 3 | −3.5e−07 | 50 |
| | 1.05 | 0 | 0 | −3.5e−07 | 10 | −3.5e−07 | 50 |
| | 1.1 | 0 | 0 | −3.5e−07 | 25 | −3.5e−07 | 50 |
| | 1.15 | 0 | 0 | −3.5e−07 | 25 | −3.5e−07 | 50 |
| | 1.2 | 0 | 0 | −3.5e−07 | 50 | −3.5e−07 | 50 |
| | 1.25 | 0 | 0 | −3.5e−07 | 50 | −3.5e−07 | 50 |
| | 1.3 | 0 | 0 | −3.5e−07 | 50 | −3.5e−07 | 50 |
| | 1.35 | 0 | 0 | −3.5e−07 | 50 | −3.5e−07 | 60 |
| | 1.4 | 0 | 0 | −3.5e−07 | 50 | −3.5e−07 | 75 |
| | 1.45 | 0 | 0 | −3.5e−07 | 50 | −3.5e−07 | 100 |
| | 1.5 | 0 | 0 | −3.5e−07 | 50 | −3.5e−07 | 100 |
| **Crop yield** | 1 | 0 | 0 | 201.5 | 201.5 | 200 | 225 |
| | 1.05 | 0 | 0 | 200.5 | 225 | 200 | 250 |
| | 1.1 | 0 | 0 | 200 | 225 | 200 | 250 |
| | 1.15 | 0 | 0 | 200 | 225 | 175 | 250 |
| | 1.2 | 0 | 0 | 200 | 250 | 175 | 250 |
| | 1.25 | 0 | 0 | 200 | 250 | 175 | 250.25 |
| | 1.3 | 0 | 0 | 181 | 250 | 175 | 251.25 |
| | 1.35 | 0 | 0 | 175 | 250 | 155 | 275 |
| | 1.4 | 0 | 0 | 175 | 250 | 151.25 | 275 |
| | 1.45 | 0 | 0 | 175 | 250.5 | 150.625 | 275 |
| | 1.5 | 0 | 0 | 175 | 251.25 | 150.25 | 300 |
| **Annual crop income** | 1 | 0 | 0 | 3585.37 | 3585.37 | 3187 | 3983.75 |
| | 1.05 | 0 | 0 | 3234.8 | 3585.38 | 3187 | 3995.7 |
| | 1.1 | 0 | 0 | 3202.93 | 3983.75 | 3187 | 4182.94 |

**Table A2.** *Cont.*

| Outcome Variables | Gamma | sig+ | sig- | t-hat+ | t-hat- | CI+ | CI- |
|---|---|---|---|---|---|---|---|
| | 1.15 | 0 | 0 | 3192.98 | 3983.75 | 2788.63 | 4382.13 |
| | 1.2 | 0 | 0 | 3187 | 3983.75 | 2788.63 | 4780.5 |
| | 1.25 | 0 | 0 | 3187 | 3991.72 | 2788.62 | 4780.5 |
| | 1.3 | 0 | 0 | 3187 | 4015.62 | 2438.06 | 4780.5 |
| | 1.35 | 0 | 0 | 2812.53 | 4382.12 | 2410.17 | 4784.48 |
| | 1.4 | 0 | 0 | 2788.63 | 4382.13 | 2402.2 | 4792.45 |
| | 1.45 | 0 | 0 | 2788.62 | 4780.5 | 2398.22 | 4812.37 |
| | 1.5 | 0 | 0 | 2788.62 | 4780.5 | 2394.23 | 5015.54 |

- Choosing a Matching Algorithm

Different matching algorithms can be used to estimate the average treatment effect. These include kernel matching, radius matching, stratified, and nearest neighborhood matching. The selection of the best matching algorithm depends on a large matched sample size, low pseudo $R^2$, large number of insignificant variables after matching, and low standardized mean bias. Therefore, the study used four matching algorithms to estimate the treatment effect (kernel, nearest neighbor, radius, and stratification matching).

**Table A3.** Performance of matching methods.

| Matching Methods | Ps R2 | LR chi2 | p > X$^2$ | Mean Bias | Med Bias |
|---|---|---|---|---|---|
| Before matching | 0.317 | 418.34 | 0.000 | 13.1 | 14.3 |
| NNM (5) | 0.006 | 4.89 | 0.998 | 4.2 | 4.1 |
| KM (bind width .06) | 0.006 | 5.62 | 0.897 | 5.3 | 6.1 |
| RM (,06) | 0.006 | 4.89 | 0.998 | 4.2 | 4.1 |
| SS | 0.006 | 4.89 | 0.998 | 4.2 | 4.1 |

Figure A1 indicates that, in the common support region, contributors and non-contributors of similar characteristics are compared to each other when estimating the ATT. The common support assumption is satisfied, because the treated and control group have been matched based on observable characteristics. The region of common support is between 0.11463415 and 0.9895831. Common support is subjectively assessed by examining a graph of propensity scores across treatment and comparison groups (Figure A2). The vertical or Y-axis in the common support represents the "density", whereas the horizontal or X-axis displays the propensity score

Figure A2 plots the distributions of propensity scores before matching. Figure A3 indicated the significance difference between treated and control groups before matching, implying the need to use matching methods to balance the distribution of the treated and control groups.

Figure A3 depicts the distribution of the propensity scores after matching using a nearest neighbor matching algorithm. The kernel density plots in Figure A4 indicate the distribution of the propensity scores after matching between treated and control groups nearly overlap and are similar, justifying the relevance of the use of nearest neighbor matching.

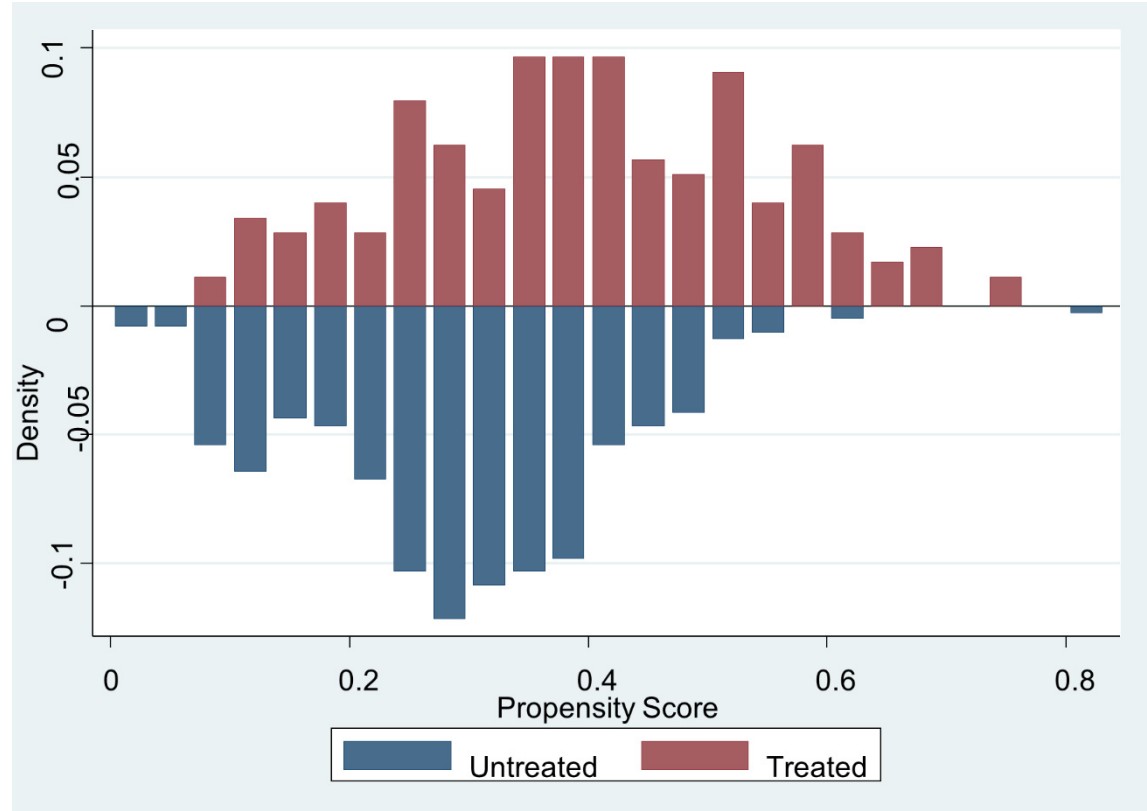

**Figure A1.** Common support region.

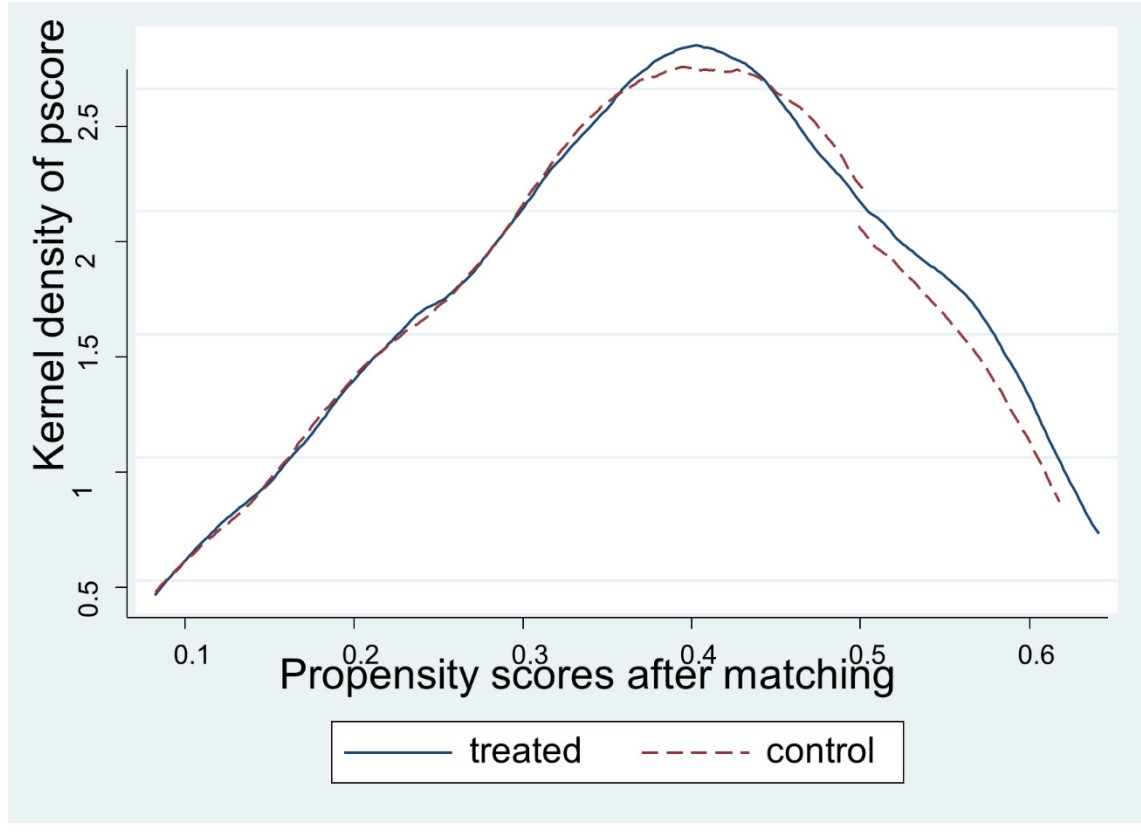

**Figure A2.** Graph of propensity score before matching.

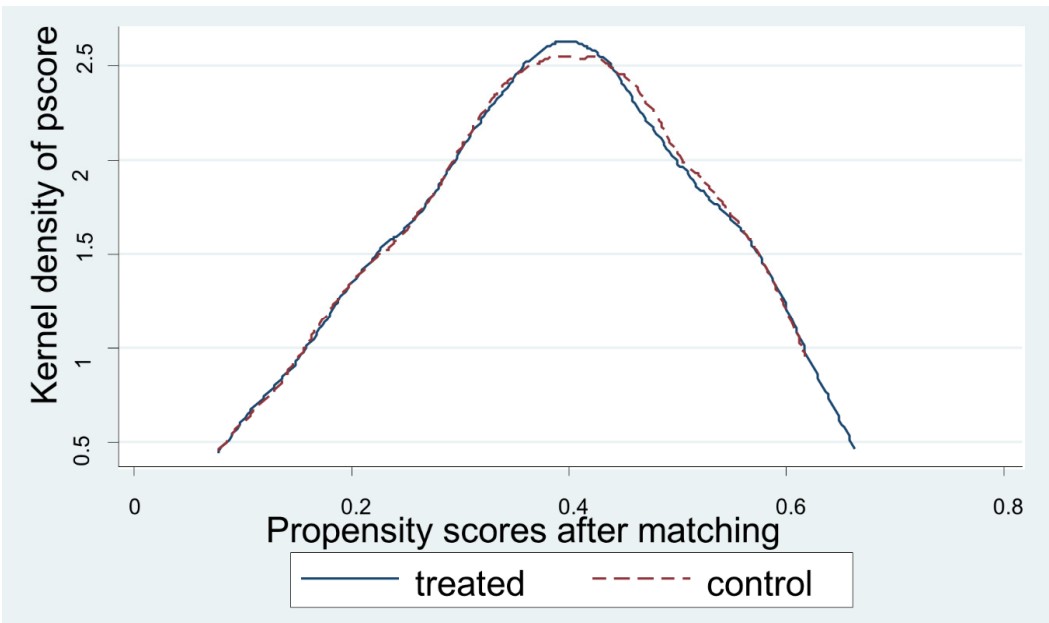

**Figure A3.** Graph of propensity score after nearest neighbor matching.

Figure A4 illustrates the distribution of the propensity scores after matching using a kernel matching algorithm. The kernel density plots in Figure A5 indicates the distribution of the propensity scores after matching between treated and control groups nearly overlap and are similar, justifying the relevance of the use of kernel matching.

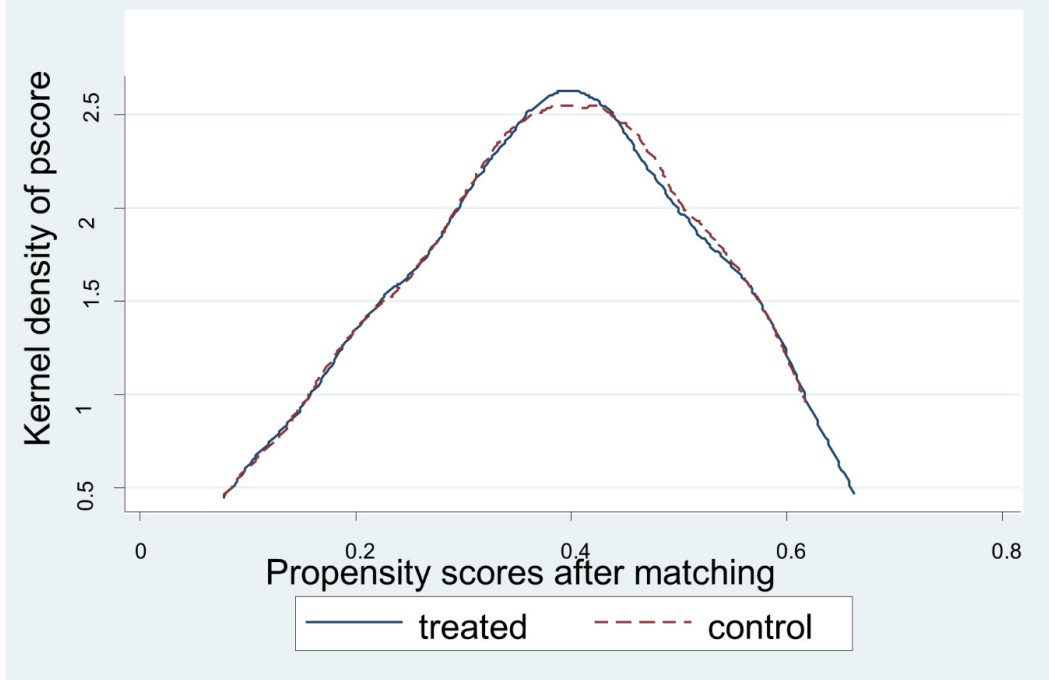

**Figure A4.** Graph of propensity score after kernel matching.

Figure A5 show the distribution of the propensity scores after matching using a radius matching algorithm. The kernel density plots in Figure A6 indicate the distribution of the propensity scores after matching between treated and control groups nearly overlap and are similar, illustrating the appropriateness of the use of radius matching.

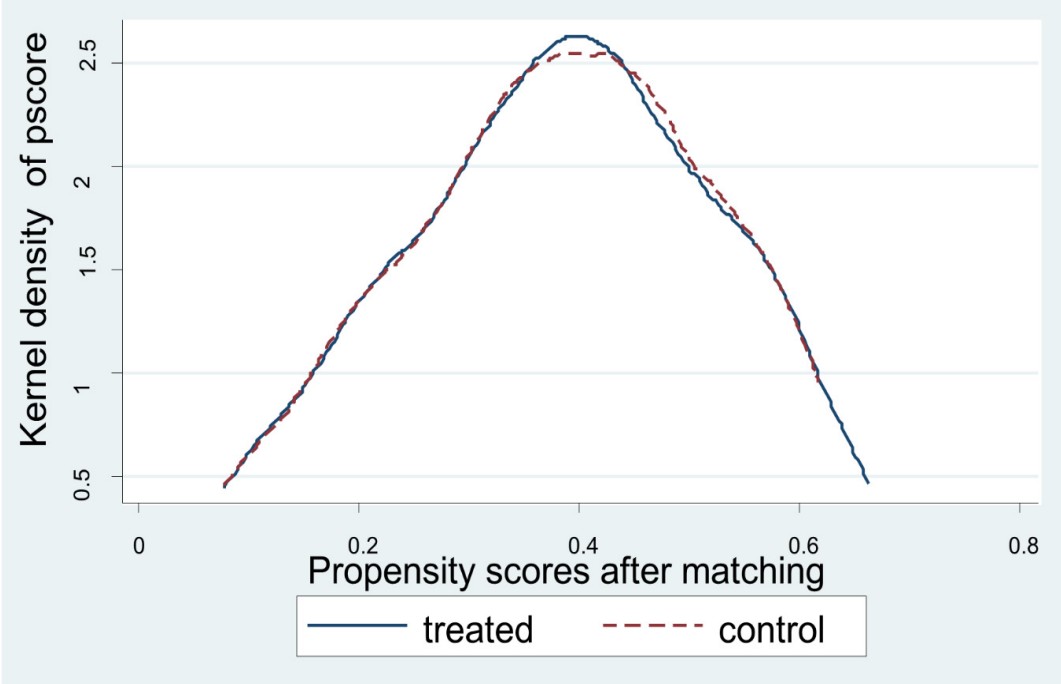

**Figure A5.** Graph of propensity score after radius matching.

Figure A6 portray the distribution of the propensity scores after matching using a stratification matching algorithm. The kernel density plots in Figure A7 indicate the distribution of the propensity scores after matching between treated and control groups nearly overlap and are similar, indicating the relevance of stratification matching methods.

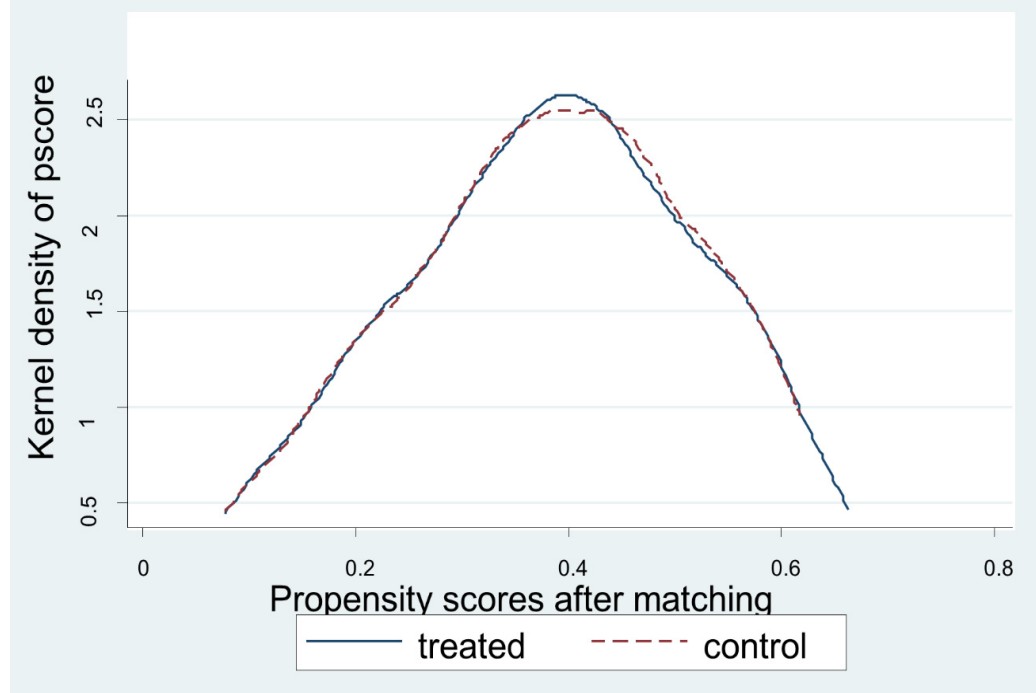

**Figure A6.** Graph of propensity score after stratification matching.

## Appendix C. Normality Test

A standardized normal probability plot was used to examine the normality of fertilizer (Figure A7), farmyard manure (Figure A8), yield (Figure A9), and income (Figure A10). Based on the figures, we observe that there is not much variability in the variables as well as outliers. Hence, the outcome variables used in this paper are normally distributed and robust for the analysis.

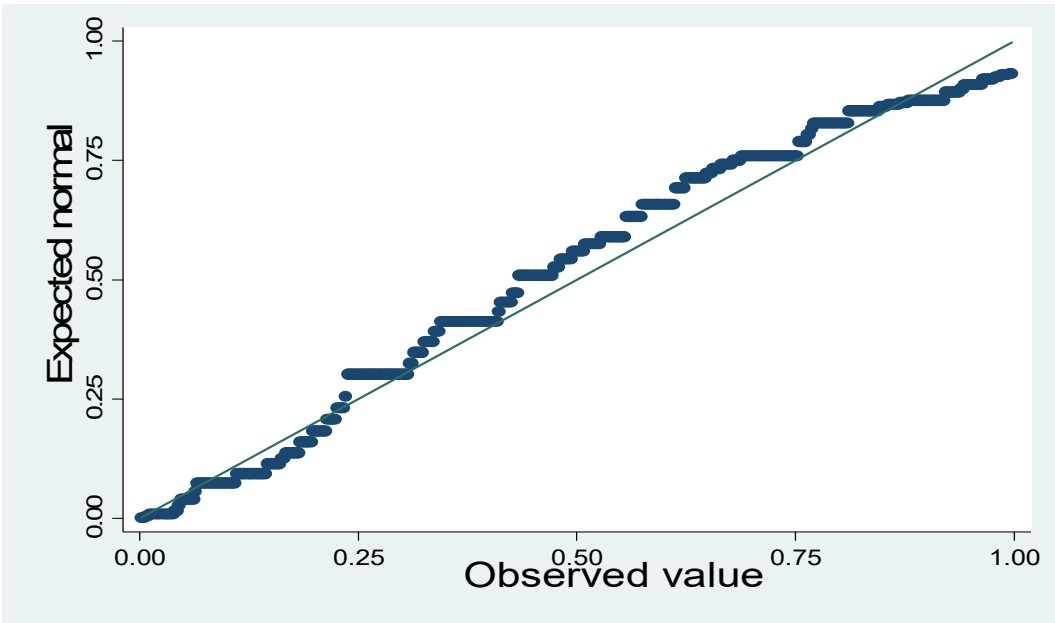

**Figure A7.** Standardized normal probability plot (P-P) of amount of fertilizer used.

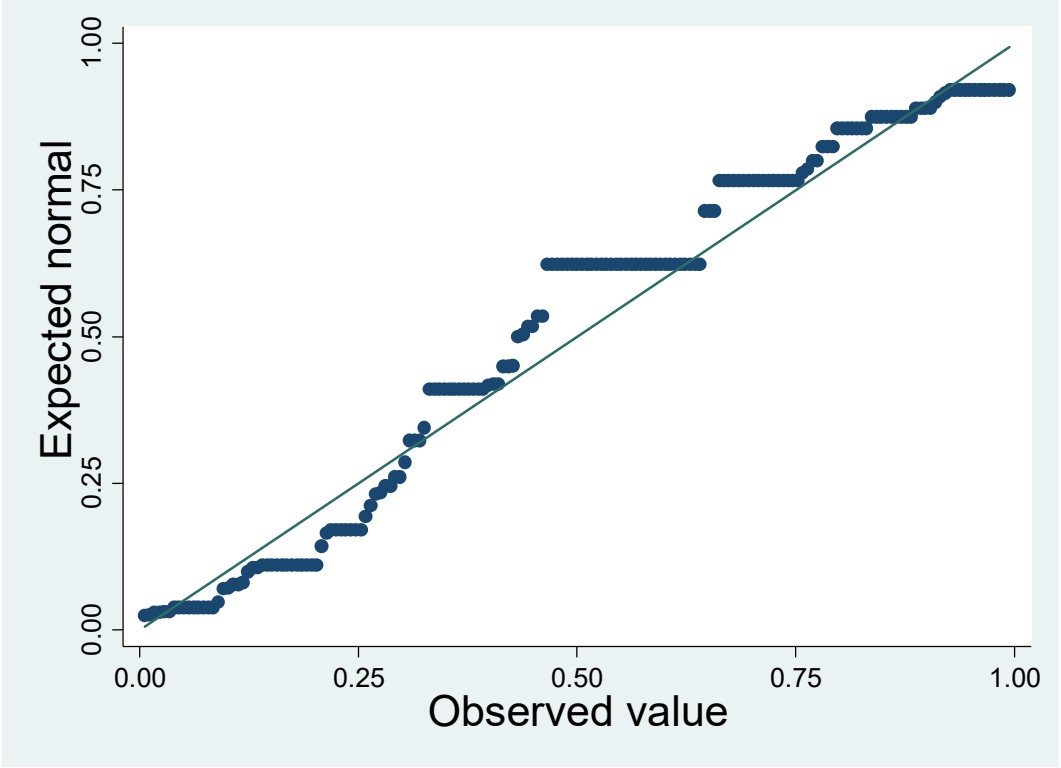

**Figure A8.** Standardized normal probability plot (P-P) of amount of farmyard manure used.

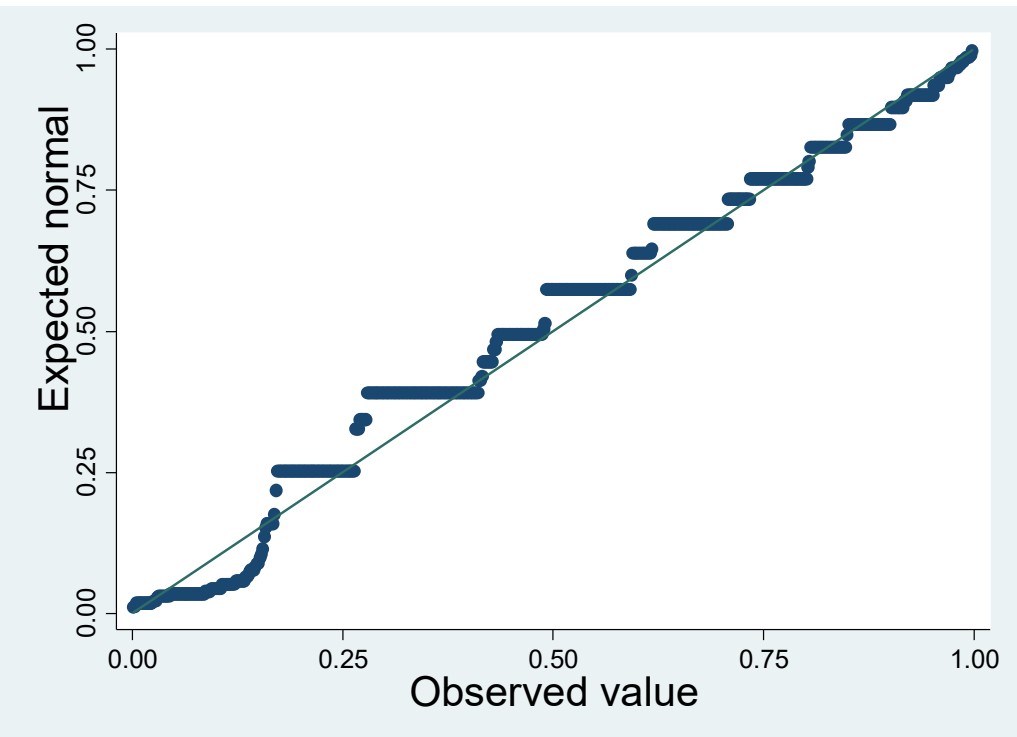

**Figure A9.** Standardized normal probability plot (P-P) of crop yield.

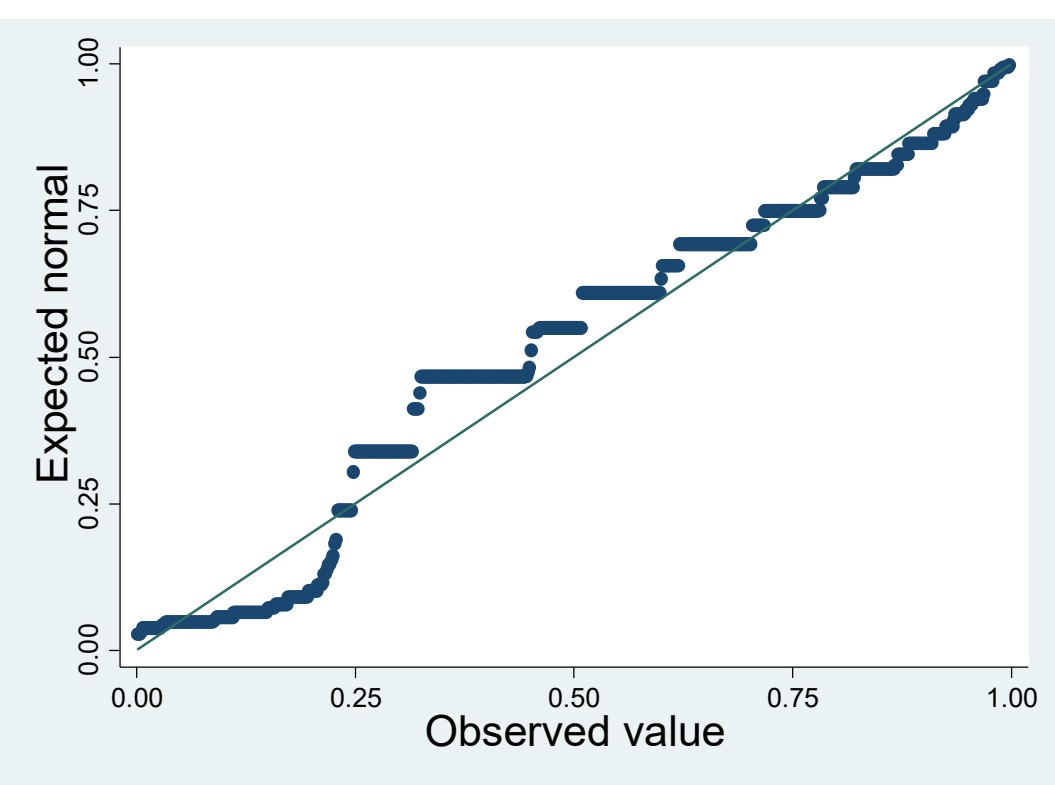

**Figure A10.** Standardized normal probability plot (P-P) for annual crop income.

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
