# Peer review of "Adoption of Road Water Harvesting Practices and Their Impacts: Evidence from a Semi-Arid Region of Ethiopia"

_sustainability, doi:10.3390/su12218914_

Round 1

Reviewer 1 Report

This is a very solid research paper on a significant practice.  The research methodology is presented very clearly.  I was not very familiar with the binary probit model and the propensity score matching methods used in this article, but nonetheless it was communicated very well, to the extent that the reader can be confident in systematic approach from start to finish.  Furthermore, the appendices gave further tables, so that the data analysis is very transparent and understandable, but yet the article is not clogged up with methodological discussion only. 

There is a clear account of the significant finding and the degree to which some factors were impactful, such as the education level of the household, family labor, access to markets, and distance of the farming plot from the farmer’s
dwelling.  

The article is extremely well organized and precisely written, but I found one  sentence on line 338-9 starting with 'policies should strive ...' that needs editing.

I recommend publishing subject to one small addition on gender analysis-

1. I would request further clarity on the number of female headed households that were included in the study? What percentage of the 159 was it? Was it large enough to provide statistical significance? Could we have more information on the this aspect given it comes into the conclusions and discussion although there is not much analysis of it in the substance of the article. I am asking because as it stands it is difficult to link the very slight gender analysis provided in the main body with the gender relevant recommendation from line 342 on where the article states : 'It requires sensitivity to the impact of different road water harvesting options for male and female livelihoods, better linkages to male/female roles in different socioeconomic contexts, ensuring female representation in local consultation processes, and consideration of special measures to engage and support female-headed households in better road water harvesting and other opportunities created by roads for resilience [19].' 

Author Response

Point 1. I would request further clarity on the number of female headed households that were included in the study? What percentage of the 159 was it? Was it large enough to provide statistical significance? Could we have more information on the this aspect given it comes into the conclusions and discussion although there is not much analysis of it in the substance of the article. I am asking because as it stands it is difficult to link the very slight gender analysis provided in the main body with the gender relevant recommendation from line 342 on where the article states : 'It requires sensitivity to the impact of different road water harvesting options for male and female livelihoods, better linkages to male/female roles in different socioeconomic contexts, ensuring female representation in local consultation processes, and consideration of special measures to engage and support female-headed households in better road water harvesting and other opportunities created by roads for resilience [19].' 

Responses.  We sincerely appreciate your suggestions to improve our work.  We acknowledge that a gender-disaggregated analysis will guide the future research and we believe this is an important research gap in the current literature. It is true that while describing the sample household gender was not indicate in the previous version though we have sufficient data to elaborate. The goal of our manuscript is to provide empirical evidences on the impacts of road water harvesting in general by taking randomly sampled households around the road networks for evaluation in general term. However, we do consider that gender disaggregated analysis for the impacts evaluation will be an important goal of our future research. As per the direction, we added additional results in Table 3 and the subsequent discussion as presented in line 278-285 based on the result we also modified the conclusion section.

Reviewer 2 Report

In this paper, the authors investigate the factors affecting on farmers’ decision to adopt RWH practices and its effects on input use, yield and farm income. Generally, the subject of the paper is interesting, but there are some points that need to be attended.  

  1. English language editing of the manuscript is required.
  2. All the abbreviations used for first time should be explained. E.g., GDP, PSM, ATT…
  3. In all Tables, titles and variables/coefficients description should be improved.
  4. Line 125: what “kebele’s farmers” means?
  5. Lines 192-193: I don’t understand the symbols in the parenthesis. Please explain
  6. Line 201: p<0.01 or p<0.1? Please check.
  7. Line 278: what the numbers in the bracket mean?
  8. In Table 4, what “ATT” means?

Author Response

Point1. English language editing of the manuscript is required.

Response. Thank you for the suggestion. English language edition was made throughout the documents according to the suggestion 

Point 2. All the abbreviations used for first time should be explained. E.g., GDP, PSM, ATT…

Response. All abbreviations used for the first time has been explained throughout the documents.

See Line 38-39 Gross Domestic Product (GDP),

See line 199. Propensity score Matching (PSM)

See line 210. Average treatment effect on treated (ATT) etc

Point 3. In all Tables, titles and variables/coefficients description should be improved.

Response : Thanks for your suggestion. In the revised versions, this has been improved throughout the document.

Point 4. Line 125: what “kebele’s farmers” means?

  • Explanation for kebele has been given in the revision manuscript. See line 186-187: In Ethiopia, kebele is the lowest administrative unit

Point 5. Lines 192-193: I don’t understand the symbols in the parenthesis. Please explain

  • The symbol x2 repents for chi-square which explains over model fitness

Point 6. Line 201: p<0.01 or p<0.1? Please check.

  • It is checked the p-value is 0.01(see table 3)

Point 7. Line 278: what the numbers in the bracket mean?

Response. Thank you for raising this issue. We explained the common support region in more detailed to make more clear in the revised version of the manuscript. See line379-387. The performance of the matching model was checked through different tests. For this, the common support region [0.04450968, and 0.76567991], which ensures that the mean propensity scores for RWHP users and non-users was selected. The common support region is a region where the values of propensity scores of both adopters and comparison groups defined. The region of common support will be defined by dropping observations below the maximum of the minimums and above the minimum of the maximums of the balancing scores between the two groups. Then the Average Treatment Effect on treated (ATT) are only determined in the region of common support.

Point 8. In Table 4, what “ATT” means?

Reponses:  Average treated effect on treated or ATT is the impact of adoption of the WHP on the adopters in comparison with the non-adopters. For this detailed explanation has been has been given from line 207-242. In Addition, we also explained what ATT means in in Table 4 of the revised version of the manuscript. 

Reviewer 3 Report

Your paper on the “Adoption of Road Water Harvesting Practices and their Impacts: Evidence from a Semi-arid Region of Ethiopia” is overall relevant, interesting, well developed and well written. I liked reading it. Nevertheless, I have a couple of issues that need to be addressed.

General issues:

(1) The region of your investigation “Tigrai” is not well introduced in section 2.1. What does “Tigrai” refer to? Is it an administrative unit? Please clarify. Please also add a map of the region and its location in Ethiopia. In addition it would be nice to show some pictures on how RWHP look like in this region. It’s difficult to imagine. Are the roads paved, for instance?

(2) The survey instrument needs better explanation in section 2.2. What kind of questionnaire was used. Standardized, semi-standardized? What was the structure of the questionnaire? Please provide more details. Please make the questionnaire (or at least the relevant parts) available for example in the supplementary material.

(3) Please mention which software you used to perform the data analysis. I guess from the Figure B1 to B5 that you used Stata. This needs to be mentioned . Further you should either make the data set available in the supplement or at least mention whether the data set data can be made available or upon request. 

(4) Please explain all abbreviations when used for the first time. In the tables, all abbreviations should be explained in the notes to the tables in order to enable the reader to understand the table stand-alone.

(5) Please add some words in the conclusion about the limitations of your study.

Specific issues

 L 95-97: Since you follow the standard structure of scientific reports, you don’t need to present the structure of the paper. These three line can be deleted.

L 143: The abbreviation ATT is not explained when used for the first time here. Please explain.

L 159: Table 1: please cluster the variables in household level and plot level confounding factors and outcome variables.

L 183: Table 2: Please cluster all variables into household characteristics and plot characteristics. Please always add the units of the variable in parenthesis. This will allow you to create a better shape of the table. For instance “Sex of household head. Female” could be rewritten as “Sex of household head. Female (%)” and could be moved to the household characteristics. The column heading “Household categories by water harvesting (mean)” could be rewritten as “Variable means by Road Water Harvesting Practice”. The number of observation for each group should be added for the household characteristics and the plot characteristics.

L 194: when you mean statistically significant you should write “statistically significant” instead of “significant”

L 241: Table 3: Please add the number of observations to the table.

L 296: Table 4: Please explain all abbreviations used in this table in the notes to the table. Such as NN, RR, KK, SS and ATT. I don’t understand the “Number of users” and “Number of non-users”. Are the observations at the household level or at the plot level. I guess at the plot level. Then it must read “Number of plots of RWHP users” and “Number of plots of RWHP non-users”. Clarify. Why do the numbers of non-users sometimes change from 138 to 404? Explain.

L 436: “…pseud R2…” must read “…pseudo R2…”

L 447: Table B1: Do the variables fulfill the distributional assumptions to perform a reliable t-test? Clarify.

Author Response

General issues:

Point 1. The region of your investigation “Tigrai” is not well introduced in section 2.1. What does “Tigrai” refer to? Is it an administrative unit? Please clarify. Please also add a map of the region and its location in Ethiopia. In addition it would be nice to show some pictures on how RWHP look like in this region. It’s difficult to imagine. Are the roads paved, for instance?

Response. Thank you for your helpful comment. To address this concern, we added brief explanation of the region biophysical, geo-politics and socio-economic descriptions (see line100-167) in the revised manuscript).

  • Map of the region and locations of water harvesting practices has also included (see line 142-147 in the revised manuscript).
  • Explanation of the road types (paved and unpaved roads) (see line152-161 in the revised manuscript).
  • Pictures on how RWHP look like in this region(See line figure 2 a, b, c, d line 161-167

Point 2. The survey instrument needs better explanation in section 2.2. What kind of questionnaire was used. Standardized, semi-standardized? What was the structure of the questionnaire? Please provide more details. Please make the questionnaire (or at least the relevant parts) available for example in the supplementary material.

    Responses.  To clarify this section, we added the statement as follow: Line 189-194:  The household-level interviews were conducted during home visits using semi-structure interview schedule. The major data set for the interview schedule includes household demographic, socio-economics, land use, use of road water harvesting, yield, input use, income and farmers’ viewpoints concerning the effects of road water harvesting and the perceived benefits and losses related to the road water harvesting practices (See supplementary material on the details of data collection tool)

Point 3. Please mention which software you used to perform the data analysis. I guess from the Figure B1 to B5 that you used Stata. This needs to be mentioned . Further you should either make the data set available in the supplement or at least mention whether the data set data can be made available or upon request. 

Responses. Thank you for your comment. To improve the clarification, we added the statement as follow: Line 252-253:  ‘Quantitative data analyses were carried out using Stata software version 14. ‘

Point4. Further indicated that data set can be made available or upon request. 

Thank you. The suggestion is well taken see line 483.

Point 5.  Please explain all abbreviations when used for the first time. In the tables, all abbreviations should be explained in the notes to the tables in order to enable the reader to understand the table stand-alone.

Responses. All abbreviations has been explained as per the direction, thanks for the direction!

Point 5. Please add some words in the conclusion about the limitations of your study.

Response. We truly appreciate your comment. Considering your valuable suggestion about the conclusion, this section has been revised as follows See line 434-470:

‘Some of the limitations of this study should be outlined. First, the data for this study were limited to one point in time; as such, there was no temporal component to the analysis. Second, this study only used quantitative approach as such some non-quantifiable qualitative variables that affect uses of RWHP and its impacts may not be fully capture. Hence, while our goal was to show the casual relationship between road water harvesting and its impacts, we recommend use of panel and mixed methods approaches as an important next step to guide future research on the relationships between RWHP and its multidimensional impacts.’

Specific issues

 L 95-97: Since you follow the standard structure of scientific reports, you don’t need to present the structure of the paper. These three line can be deleted.

Response. Thanks for your suggestion. This has been deleted in the revised manuscript

L 143: The abbreviation ATT is not explained when used for the first time here. Please explain.

Response. Thanks for your suggestion. This has been well taken throughout the document

L 159: Table 1: please cluster the variables in household level and plot level confounding factors and outcome variables.

Response. Thank you for your comment. This has been well taken throughout the document

L 183: Table 2: Please cluster all variables into household characteristics and plot characteristics. Please always add the units of the variable in parenthesis. This will allow you to create a better shape of the table. For instance “Sex of household head. Female” could be rewritten as “Sex of household head. Female (%)” and could be moved to the household characteristics. The column heading “Household categories by water harvesting (mean)” could be rewritten as “Variable means by Road Water Harvesting Practice”. The number of observation for each group should be added for the household characteristics and the plot characteristics.

Response. Thank you for your remark.

  • Variable cluster has been made throughout the document for the household characteristics and the plot characteristics(see Table 1, Table 2 and Table 3)
  • Column heading has been modified according to the direction ( “Variable means by Road Water Harvesting Practice)
  • The Number of observation has been added on each Table heading

L 194: when you mean statistically significant you should write “statistically significant” instead of “significant”

Response: Thank you for your remark.

  • Thank you for your comment. This has been well taken throughout the document

L 241: Table 3: Please add the number of observations to the table.

Responses. Number of observation has been added in each Table heading as per the direction

L 296: Table 4: Please explain all abbreviations used in this table in the notes to the table. Such as NN, RR, KK, SS and ATT. I don’t understand the “Number of users” and “Number of non-users”. Are the observations at the household level or at the plot level. I guess at the plot level. Then it must read “Number of plots of RWHP users” and “Number of plots of RWHP non-users”. Clarify. Why do the numbers of non-users sometimes change from 138 to 404? Explain.

Response. Thank you for the suggestion. As balancing property, PSM often matches observations with similar characteristic (i.e household and plot in this paper). While balancing some of the observation particular that of the non-participants in road water harvesting practices could be excluded during the analysis systematically which makes sample observation to vary by the different algorism as indicated in the methodology section. Sample size of different Matching algorithms can be vary depend on type of Matching algorithms. In this study, we used different Matching algorithms to estimate the ATT. Accordingly; the sample size of matching algorithms was different. For example in our study we use nearest neighbour matching and the sample size is low as compare to others because nearest neighbour matching estimates ATT by comparing treated individual in terms of the closest propensity score (participants with similar pscore). However, nearest neighbour with calliper matching was used to reduce the bias. Radius matching also other matching method based on comparison group members within a given radius. In radius matching participant far from a given radius, value is excluded from analysis. Kernel matching also nonparametric matching estimators that compare the outcome of each treated individuals to a weighted average of the outcomes of the entire control group and stratification is one of PSM matching estimator based on strata formation. Therefore, in stratification first participants are ranked based on pscore and dived into number of strata to estimate ATT. Therefore, the sample size of different Matching algorithms is different due to the above case. However, the selection of matching algorithms was performed based on the selection criteria which are low and insignificant pseud R2, low mean bias and insignificant covariates after matching.  To make this issue more clearly we added a pararagaphy (see from Line 214-242).

Point L 436: “…pseud R2…” must read “…pseudo R2…”

Responses. Comment well taken

Point L 447: Table B1: Do the variables fulfil the distributional assumptions to perform a reliable t-test? Clarify.

Responses.  Thank you for the direction. PSM is non-parametric statistics that has its own assumption as indicated in the analytical producer sections of the material methods chapter. Our data has fulfilled all the assumption of PSM as indicated in the annexation section and as discussed in each table.  In addition to this we also conducted box plot and shipro Wilk Normality Test, the result indicate the overall model predication was normality distributed. For detail of normality test kindly see line 586-599.

Reviewer 4 Report

The manuscript presents an interesting study and should be published after major revision.

 The comments and required clarifications are listed below:

  • English should be corrected in some parts of text e.g. line 16: potential has yet not been…; line 53: changes have been recently experienced…; lines 53-55: the word "serious" is repeated in the sentence; line 235: the order of the sentence is wrong;
  • Lines 95-97 can be deleted;
  • Lines 44-52, please give some data, e.g. range of rainfall variability in Ethiopia;
  • Point 2.1, please give some precise information about location, area etc.
  • Point 3.1. Are the discussed differences between the mean values statistically significant?
  • Point 3.3. If the student's t-test was used, how the normality of the tested variables was checked?
  • Please use x2 instead of chi or x2 (line 193, table 3),
  • The conclusion must be rewritten. It is more of a summary and therefore it is much too long. I recommend that you only include the take-home message, i.e. the key findings of the study, in the conclusion.
  • Please use one font format in the description of the drawings (Figures B1-B5) and provide descriptions on the vertical axes.

Author Response

 The comments and required clarifications are listed below:

Point 1. English should be corrected in some parts of text e.g. line 16: potential has yet not been…; line 53: changes have been recently experienced…; lines 53-55: the word "serious" is repeated in the sentence; line 235: the order of the sentence is wrong.

Responses. Thank you for the insight. We edited the language problem as per the direction.

This has been fixed in the revised manuscript as indicated from line 16-17… ‘In the drylands of Ethiopia, several road water harvesting practices (RWHP) have been used to supplement rain-fed agriculture. However, factors affecting adoption of RWHP and their impacts were not studied systematically.’

Lne 53: changes have been recently experienced…;…; lines 53-55: the word "serious" is repeated in the sentence;

Response. This has been fixed now. See line 53-55

line 235: the order of the sentence is wrong

Response. This has been fixed now: Kindly see line 235 of the revised manuscript. In this study, years of schooling is found to positively influence the adoption of RWH technology

Point 2. Lines 95-97 can be deleted;

Responses. Thanks for your suggestion. This has been deleted in the revised manuscript

  • Lines 44-52, please give some data, e.g. range of rainfall variability in Ethiopia;

To address this concern, we added brief explanation of the region biophysical, geo-politics and socio-economic descriptions (see line100-167) in the revised manuscript).

  • Map of the region and locations of water harvesting practices has also included (see line 142-147 in the revised manuscript).

Point 2.1, please give some precise information about location, area etc.

Responses. This has been well addressed. Location of the study area and road network has been in dictate in figure 1

  • Point 3.1. Are the discussed differences between the mean values statistically significant?

Responses. Yes, the discussion for the impact evaluation result are based on statistical significant mean values at p<5%.

  • Point 3.3. If the student's t-test was used, how the normality of the tested variables was checked?

Responses. Thank you for the direction. We have not used student t-test to predict the impact of road water harvesting rather we used PSM. For this paper, we have use t-test only for sensitivity analysis as post-estimation of propensity score match (PSM). PSM is non-parametric statistics that has its own assumption as indicated in the analytical producer sections of the material methods chapter. Our data has fulfilled all the assumption of PSM as indicated in the annexation section and as discussed in each table.  In addition to this we also conducted box plot and shipro Wilk Normality Test, the result indicate the overall model predication was normality distributed. For detail of normality test kindly see line 586-599.

  • Please use xinstead of chi or x2 (line 193, table 3),

Responses. Comment well taken

  • The conclusion must be rewritten. It is more of a summary and therefore it is much too long. I recommend that you only include the take-home message, i.e. the key findings of the study, in the conclusion.

Responses. Comment well taken. The key issues per se has been included in the revised versions of the conclusion

  • Please use one font format in the description of the drawings (Figures B1-B5) and provide descriptions on the vertical axes.

Responses. Comment well taken

Round 2

Reviewer 3 Report

Thank you very much for the revision of your manuscript which definitely improved its quality so far. However, still some major and minor issues remain.

Major issues

  1. The variables of Table 1, Table 2, Table 4, and Table 6 should be labelled consistent. Currently the labeling is often confusing: Table 1 “Distance from home to market” as a plot characteristic (is this correct?), Table 2: “Market distance from woreda (district) markets”, as a plot characteristic (is this correct?), Table 3 “Distance to woreda market”, as a household characteristic (is this correct?), Table 6 “Woreda distance from home” as a household characteristic. Very confusing. Table 1 should include all relevant variables and should divide them in household and plot characteristics. Exactly the same variable names (or abbreviations) must be repeated in all other tables! You need to be perfectly consistent!  
  2. The normality test in the Appendix C is not convincing. First, you don’t need a normality test for the independent variables. Second, the box-plots as they are right now are not useful because of the very different scales of the variables. The focus should be on the outcome variables, they are tested for differences as reported in Table 5. You need to develop arguments why you use a t-test (a parametric test) instead of a non-parametric test (for example median test). Of course, I know that Stata produces the kind of outputs that you reported and that’s fine. If some assumptions in the distribution of variables are violated you may argue for using bootstrapping, that’s possible with Stata. Please clarify and provide arguments (with appropriate references) and add them in section 3.2.

Minor issues

Line 98: Please add at the end of introduction one or two sentences about the possible relevance of your study beyond the Ethiopian context.

Line 106: “...in Gode…” Gode is a city, isn’t it? If yes it might be better to write “...in Gode city…”

Line 208: “...Regional State…” must read “...Regional States…”. “Tigrai Region founds…” might better read “Tigrai Region is located…”

Line 110: “Trigai lies in northern Ethiopia…”  In order to avoid repetition I suggest writing “It is…”

Line 129: “for cultivation.” Please provide a reference.

Line 244: “...Propensity score Matching…” should better read “...Propensity Score Matching…”  

Line 255, Table 1: Some variables are missing in this table (e.g. Amount of farmyard manure used, Literacy rate), please check carefully. This table must include all relevant variables. Does “Distance from home to market” really belong to plot characteristics? “Input use” must be “Amount of fertilizer used”. Check for consistency.

Line 279; Table 2: As I already mentioned in my first review. The units should be given for all variables in parenthesis, e.g. “Sex of household head Female (%)”, “Male (%)”, “Access to credit (%)”, Annual crop income (Ethiopian Birr), etc.  Then you can drop the “percent” in the row above Sex of households. Be consistent with the variable names. Is it true that on average the schooling of the household head is just 1.38 or 2.56 years? Very low. Are the numbers correct?  Please express the numbers in Ethiopian Birr also in $ values of a specific year in a note to the table.

Line 554: What is the variable input use? Usually the variables “Fertilizer” and “Manure” were used. Clarify.

Line 571: Figure B2. The unit for the vertical axis is missing. Same for Figure B3, B4, B5, B6.

Author Response

Major issues

Major point 1. The variables of Table 1, Table 2, Table 4, and Table 6 should be labelled consistent. Currently the labeling is often confusing: Table 1 “Distance from home to market” as a plot characteristic (is this correct?), Table 2: “Market distance from woreda (district) markets”, as a plot characteristic (is this correct?), Table 3 “Distance to woreda market”, as a household characteristic (is this correct?), Table 6 “Woreda distance from home” as a household characteristic. Very confusing. Table 1 should include all relevant variables and should divide them in household and plot characteristics. Exactly the same variable names (or abbreviations) must be repeated in all other tables! You need to be perfectly consistent! 

Response. Thank you so much for critically identifying the inconsistent points. In the revised version all variables in the Table has been and clustered. Yes, it is true we have some times put Distance from home to market as plot on other time as household characteristics. Our apology for the inconsistency in use of the variables! Now this variable has been kept under household characteristics and made consistent across all the tables. Besides, other variables such as literacy, productivity,  amount  of fertilizer used, amount of farm yard manure used (which were not included in Table 1 previously) has been all included now in the revised versions of the manuscript.

Major point 2. The normality test in the Appendix C is not convincing. First, you don’t need a normality test for the independent variables. Second, the box-plots as they are right now are not useful because of the very different scales of the variables. The focus should be on the outcome variables, they are tested for differences as reported in Table 5. You need to develop arguments why you use a t-test (a parametric test) instead of a non-parametric test (for example median test). Of course, I know that Stata produces the kind of outputs that you reported and that’s fine. If some assumptions in the distribution of variables are violated you may argue for using bootstrapping, that’s possible with Stata. Please clarify and provide arguments (with appropriate references) and add them in section 3.2.

Response. We truly appreciate your comment. Considering your valuable suggestion about the normality test we exclude the normality test of the independent variables.  For the outcome variables (eg. Farm yard manure use, fertilizer use, yield and income) we also develop independent normality test graphs using standardized normal plot. These types of normality test are widely used this days. The normality test used in this study also justifies data are normally distributed and appropriate for drawing plausible conclusion.

For the normality test, the following explanation was also given in line 259-273…. ‘Previous authors have suggested the need for normality test in statistical analysis of numeric dependent variables. Normality can be tested either by graphical or numerical approaches. The numeric approaches are useful for making objective judgment of normality but less sensitive for small sample size or overly sensitive for large sample sizes, in this case graphical test of normality is more preferable [39]. There are numerous graphic methods to test the normality continuous data. The well-known graphic normality tests includes P–P Plot (widely known as probability- probability plot or standardized plots), box plot, Q–Q Plot (quantile-quantile Plot), Shapiro–Wilk test, Kolmogorov–Smirnov test and histograms[40]. As compared to other graphic normality tests, P-P plots is more precise for large data set. A P–P plot is a probability plot for assessing how closely fit the expected and observed value of a given data sets. When the data sets is normality distributed, the P-P plots becomes approximately straight line. Data far apart from this straight line indicates the existences of outliers and lack of normality in the data set. By visualizing the P-P plot, one can make decision about outlier, skewness, kurtosis and hence this method of normality test has become a very popular tool for testing the normality assumption [41]. Following these extra advantages, its practical simplicity and strengths in applied research, the P-P plot normality test was used in this paper.’

For the use of mean based t- test estimation procedure the following justification has also given in the text(line 306-317): Various statistical methods used for data analysis make assumptions about normality, including correlation, regression, t‑tests, and analysis of variance. If a continuous data follow normal distribution, then it is recommended to present such data in mean values. These values are further used to compare between the groups by calculating the significance level. If data are not normally distributed, resultant mean will not be a representative value of the data set. A wrong selection of the representative value of data set and further calculated significance level using this representative value might give wrong interpretation[39]. Following this background, in this paper, first we test the normality of the data then we checked whether mean is applicable as representative value of the data or not. Once mean is found applicable, RWHP users and non-users group mean was compared using parametric test otherwise, medians would have been used to compare the groups, using nonparametric methods. Such approach has been widely used by several authors in impact evaluation researches[33, 42, 43,44,45].

Minor issues

Line 98: Please add at the end of introduction one or two sentences about the possible relevance of your study beyond the Ethiopian context.

Response. Thank you. We have added the following sentence.

Line 97-1000: ‘The evidences generated from this paper would help policy makers in the dryland areas of the world particularly in sub Saharan African countries to formulate and promote appropriate road water harvesting strategies in the future as the demand for water in the era of climate change is quite pertinent.’

 Line 106: “...in Gode…” Gode is a city, isn’t it? If yes it might be better to write “...in Gode city…”

Response. Comment well taken (Kindly see line 107 in the revised versions of the manuscript)

Line 208: “...Regional State…” must read “...Regional States…”. “Tigrai Region founds…” might better read “Tigrai Region is located…”

Response. Comments well taken: (Kindly see line 110 in the revised versions of the manuscript)

Line 110: “Trigai lies in northern Ethiopia…”  In order to avoid repetition I suggest writing “It is…”

 Response. Comments well taken: Kindly see line 111 in the revised versions of the manuscript

Line 129: “for cultivation.” Please provide a reference.

. Comments well taken: Kindly see line 131 in the revised versions of the manuscript

Line 244: “...Propensity score Matching…” should better read “...Propensity Score Matching…”  

Response. Comment well taken: Kindly see line 202 in the revised versions of the manuscript

Line 255, Table 1: Some variables are missing in this table (e.g. Amount of farmyard manure used, Literacy rate), please check carefully. This table must include all relevant variables. Does “Distance from home to market” really belong to plot characteristics? “Input use” must be “Amount of fertilizer used”. Check for consistency.

Response. Thank you for the helpful insight. In the revised version of the manuscript in Table 1 all relevant variables used in the paper has been included and double checked for consistency throughout the document.

Response. It is true distance from home to market is household variable. This has been fixed now in the revised version of the manuscript. 

Line 279; Table 2: As I already mentioned in my first review. The units should be given for all variables in parenthesis, e.g. “Sex of household head Female (%)”, “Male (%)”, “Access to credit (%)”, Annual crop income (Ethiopian Birr), etc.  Then you can drop the “percent” in the row above Sex of households. Be consistent with the variable names. Is it true that on average the schooling of the household head is just 1.38 or 2.56 years? Very low. Are the numbers correct?  Please express the numbers in Ethiopian Birr also in $ values of a specific year in a note to the table.

 Thank for the inputs:

  • The unit for each variables has been given in a bracket and the percent in the row above sex of households has been deleted. See table 2
  • Variable naming has been checked and are consistent now
  • Yes, it is true the year of schooling of the household head was 1.38 or 2.56 years. In Ethiopia Education has been expanded very recently as a result most of the old generation are formally illiterate

Line 554: What is the variable input use? Usually the variables “Fertilizer” and “Manure” were used. Clarify.

Response. Thanks for your suggestion. Yes, it was wrongly reported as input use. In the revised version of the manuscript, we have reported for ‘’farm yard manure’’ and ‘’fertilizer” “separately.

Line 571: Figure B2. The unit for the vertical axis is missing. Same for Figure B3, B4, B5, B6.

Response. Thank you for the suggestion. In the revised manuscript we have labelled the vertical and horizontal axis for all figures. We also improved the quality of all figures (kindly see Figure B2- B7 of the revised manuscript).

Reviewer 4 Report

The explanations regarding the statistical research are appropriate. Please include them in the text, point 3.1, 3.3 respectively.

Please use χ2 instead of chi (Tab. 4, Tab. B3).

The vertical axes in the drawings are still not described. Please put descriptions on the vertical axes (Figures B1-B5).

Author Response

Point 1. The explanations regarding the statistical research are appropriate. Please include them in the text, point 3.1, 3.3 respectively.

We appreciate the helpful suggestion. We added explanation for the statistics in different section of the result and discussion chapter in the revised manuscript as indicated by the reviewer

  • For the normality test, kindly see line 259-273 in the revised version of the manuscript
  • For the use of mean based t- test estimation kindly see line 306-317 in the revised versions of the manuscript
  • For the impact evaluation kindly see line 401-419 in the revised version of the manuscript

 Point 2. Please use χinstead of chi (Tab. 4, Tab. B3).

Response. In the revised manuscript, use use χinstead of chi(see table 4, Table B3) as indicated by the reviewer

Point 3. The vertical axes in the drawings are still not described. Please put descriptions on the vertical axes (Figures B1-B5).

Response. In the revised manuscript, we added the value of the axis and description as indicated by the reviewer (see all figures in the appendix figures B and C) 

Round 3

Reviewer 3 Report

Thank you very much for the revisions which are made almost to my full satisfaction. There are still a number of inconsistencies left which should be eliminated. 

Table 1: "plot(land size)" should be changed to "Plot size",  "Yield" should be changed to "Crop yield", "Income" should be changed to "Annual crop income" in order to be consistent with Table 2, etc.  Why is the variable "Access to harvesting training" still missing in Table 1? "Number of plots owned" should be a household characteristic and not a plot characteristic! Are all outcome characteristics at the plot level? If yes, it should be clearly indicated. "Participation status in road harvesting" might clearer read ""Participation status of plot in road harvesting". "1 if the household participant in the use of RWHP..."  might be clearer as "1 if the household participates with the plot in the use of RWHP...". "Total income obtained from crop sell..." Is this per plot? Clarify! "Amount of kilograms of inorganic fertilizer used" should read ""Amount of kilograms of inorganic fertilizer used per plot" "Amount of kilograms of farm yard manure used" should read "Amount of kilograms of farm yard manure used per plot" Please also use also the same sequence of variables in the tables. It makes it much easier for the reader to follow. 

Table 2: Why are the variables "Use of improved seed" and "Land tenure status" missing in Table 2? Please add. "Education level of household heads (schooling)" must read "Education level of household head (years of schooling)". "Annual crop income (Ethiopian Birr)" seems to be incomplete and might be "Annual crop income (Ethiopian Birr/plot)"? Please check and correct. "Plot size(tsimad)" must read "Plot size (timad)". "Number of plots owned" is not a plot but a household characteristic!

Table 4: "Distance from home to market" must read "Distance from home to district market". Please keep the order of the variables.

Table 5: "Fertilizers used" should be "Amount of fertilizers used", "Farm yard manure use" should be "Amount of farm yard manure used". Yield" should be changed to "Crop yield", "Income" should be changed to "Annual crop income"

Table 6: Please keep the order of the variables. "Distance from home to market" should read "Distance from home to district market". "Year of schooling" should read "Education level of household head". 

Table B2: Please use the same sequence of variables as before. 1. Fertilizer, 2. Manure, 3. Yield, 4. Income. Furthermore: "Fertilizer" should be "Amount of fertilizers used", "Farm yard manure" should be "Amount of farm yard manure used". Yield" should be changed to "Crop yield", "Income" should be changed to "Annual crop income." 

Figure C8: "Fertilizer use" should be "amount of fertilizers used"

Figure C9: ""Farm yard manure" should be "amount of farm yard manure used"

Figure C11: "income" should be changed to "annual crop income."

Author Response

Point1. Table 1: "plot(land size)" should be changed to "Plot size",  "Yield" should be changed to "Crop yield", "Income" should be changed to "Annual crop income" in order to be consistent with Table 2, etc.  Why is the variable "Access to harvesting training" still missing in Table 1? "Number of plots owned" should be a household characteristic and not a plot characteristic! Are all outcome characteristics at the plot level? If yes, it should be clearly indicated. "Participation status in road harvesting" might clearer read ""Participation status of plot in road harvesting". "1 if the household participant in the use of RWHP..."  might be clearer as "1 if the household participates with the plot in the use of RWHP...". "Total income obtained from crop sell..." Is this per plot? Clarify! "Amount of kilograms of inorganic fertilizer used" should read ""Amount of kilograms of inorganic fertilizer used per plot" "Amount of kilograms of farm yard manure used" should read "Amount of kilograms of farm yard manure used per plot" Please also use also the same sequence of variables in the tables. It makes it much easier for the reader to follow. 

Response. Thank you so much for the critical insight. We are glad to have such reviewer. Table 1 has been modified as per the suggestion.(Kindly see line 257) 

  • All variable description, naming and rearrangement on (household/plot) variable characteristic have been modified as per the suggestion in all tables. Kindly see Table 1 in the revised manuscript for this specific point.
  • All variables order has been fixed in all tables and figures as per the sequence in Table 1(kindly see Table 1, Table 2, Table 4 and Table 6 in the revised manuscript) and all figures.
  •  Explanation that indicates all outcome variable are plot level has been given in line 216 and in each variable the word ''per plot'' included in the revised versions of the manuscript
  • Access to water harvesting training variable has been included in Table 1. 

Point 2. Table 2: Why are the variables "Use of improved seed" and "Land tenure status" missing in Table 2? Please add. "Education level of household heads (schooling)" must read "Education level of household head (years of schooling)". "Annual crop income (Ethiopian Birr)" seems to be incomplete and might be "Annual crop income (Ethiopian Birr/plot)"? Please check and correct. "Plot size(tsimad)" must read "Plot size (timad)". "Number of plots owned" is not a plot but a household characteristic!

Response. Sorry for missing including this two variables, may be wrongly deleted in the process. Use of improved seed and land tenure status has been included in Table 2 in the revised version of the manuscript.  All variables' description and rearranged has been made as per the suggestion. Kindly see Table 2 of the revised manuscript. 

Point3.  Table 4: "Distance from home to market" must read "Distance from home to district market". Please keep the order of the variables.

Response. Comment well taken. Kindly see Table 4 in the revised manuscript. 

Point 4. Table 5: "Fertilizers used" should be "Amount of fertilizers used", "Farm yard manure use" should be "Amount of farm yard manure used". Yield" should be changed to "Crop yield", "Income" should be changed to "Annual crop income"

Response. This has been fixed now. Kindly see Table 5 in the revised manuscript. 

Point 5. Table 6: Please keep the order of the variables. "Distance from home to market" should read "Distance from home to district market". "Year of schooling" should read "Education level of household head". 

Response. Variable order has been rearranged. Variable description has been modified as per the suggestion. 

Point 6. Table B2: Please use the same sequence of variables as before. 1. Fertilizer, 2. Manure, 3. Yield, 4. Income. Furthermore: "Fertilizer" should be "Amount of fertilizers used", "Farm yard manure" should be "Amount of farm yard manure used". Yield" should be changed to "Crop yield", "Income" should be changed to "Annual crop income." 

Response. All variable sequence has been modified as per the direction. 

Point 7. Figure C8: "Fertilizer use" should be "amount of fertilizers used"

Response. Comment well taken 

Point 8. Figure C9: ""Farm yard manure" should be "amount of farm yard manure used"

Response. Comment well taken 

Point 9. Figure C11: "income" should be changed to "annual crop income."

Response. Comment well taken 
